# Text-Aware Real-World Image Super-Resolution via Diffusion Model with Joint Segmentation Decoders

**Qiming Hu**[1,2,*] **Linlong Fan**[2,*] **Yiyan Luo**[2], **Yuhang Yu**[2], **Xiaojie Guo**[1,†] **Qingnan Fan**[2,†]

[1]College of Intelligence and Computing, Tianjin University

[2]vivo Mobile Communication Co. Ltd

`huqiming@tju.edu.cn` `fanlinlong703@163.com` `luoxiaohei333@gmail.com`
`yuyuhang@vivo.com` `xj.max.guo@gmail.com` `fqnchina@gmail.com`

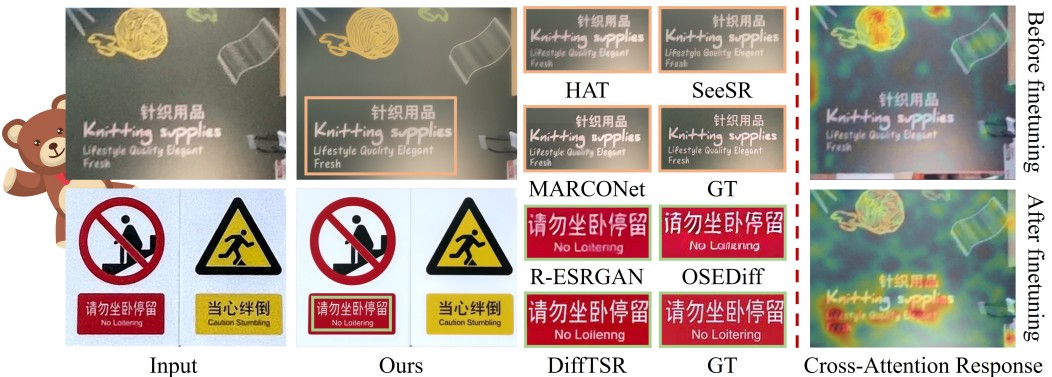

Figure 1: Left: visual comparison between SR results on real-world samples. Our method shows a clear advantage in restoring text structures over previous GAN-based [32, 4], diffusion-based [35, 34], and text image SR [12, 46] methods. Right: cross-attention to the word "text" before and after LoRA fine-tuning with joint segmentation decoders, showing improved ability to perceive text regions.

## Abstract

The introduction of generative models has significantly advanced image super-resolution (SR) in handling real-world degradations. However, they often incur fidelity-related issues, particularly distorting textual structures. In this paper, we introduce a novel diffusion-based SR framework, namely TADiSR, which integrates text-aware attention and joint segmentation decoders to recover not only natural details but also the structural fidelity of text regions in degraded real-world images. Moreover, we propose a complete pipeline for synthesizing high-quality images with fine-grained full-image text masks, combining realistic foreground text regions with detailed background content. Extensive experiments demonstrate that our approach substantially enhances text legibility in super-resolved images, achieving state-of-the-art performance across multiple evaluation metrics and exhibiting strong generalization to real-world scenarios. Our code is available at here.

## 1 Introduction

Real-world image super-resolution (Real-SR) is a challenging task that aims to recover high-resolution (HR) images from low-resolution (LR) inputs affected by unknown and coupled degradation factors

---

[*]Equal contribution.

[†]Corresponding authors.

This work was completed during an internship at vivo.

39th Conference on Neural Information Processing Systems (NeurIPS 2025).

encountered in real scenarios. In recent years, generative models such as Generative Adversarial Networks (GANs) [32, 14, 21, 4] and Diffusion Models [42, 29, 39, 35, 41, 16, 34] have been introduced into Real-SR, leveraging generative priors to hallucinate missing details lost due to degradation. Generative models can produce visually realistic images, but often sacrifice structural accuracy for perceptual quality, raising concerns about the trade-off between fidelity and realism [17]. One of the most prominent issues arises in textual content, particularly in languages with complex stroke structures such as Chinese [12]. As illustrated on the left side of Fig. 1, previous generative models often struggle to perceive and preserve textual structure in reconstructed images, resulting in severe distortions such as malformed strokes or incorrect characters. These issues not only degrade user experience but also hinder downstream applications that depend on accurate text restoration.

To accurately restore text structures degraded in real-world images, we propose to fine-tune the cross-attention mechanism between text and image tokens in a pre-trained diffusion model. As shown in Fig. 1 (right), we visualize the cross-attention response to the word "text" using DAAM [25] and observe that the original model fails to attend properly to text regions. However, by introducing a LoRA-based fine-tuning strategy [11] with a joint image super-resolution and text segmentation task, the model learns to focus its cross-attention on textual areas. These cross-attention maps, after a linear projection, are fed into a dedicated text segmentation decoder to produce high-quality text masks. This observation leads to two key insights: (1) LoRA fine-tuning can effectively guide the cross-attention of diffusion models toward previously under-attended semantic categories; and (2) text-aware super-resolution and text segmentation are highly complementary and can be unified in a multi-task learning framework. Based on these insights, we propose TADiSR (Text-Aware Diffusion model for real-world image Super-Resolution), which replaces the standard VAE image decoder with a pair of image and text segmentation decoders. The decoders take as input the cross-attention response together with the denoised latents, conducting image-segmentation interactions through a multi-scale, dual-stream manner. TADiSR can significantly enhance the fidelity of text regions in super-resolved images while preserving general image quality. Unlike prior text SR methods [19, 12, 46] that require a multi-stage process of detection, regional SR, and fusion, our approach is end-to-end, producing the final output in a single pass for better practicality.

Based on the above analysis, a dataset that contains both accurate text segmentation masks and high-quality background scenes is essential for advancing text-aware image super-resolution. However, no existing dataset fully satisfies these requirements. Existing text segmentation datasets, such as TextSeg [37] and BTS [38], mainly contain cropped text regions with limited background detail and possible degradation, and most are restricted to English, except for the bilingual BTS. General-purpose super-resolution datasets such as DIV2K [1], Flicker2K [27], and LSDIR [13] offer rich image details but contain limited text content and lack any form of text region annotation. Scene text image super-resolution datasets like Real-CE [18] typically provide both low-resolution and high-resolution image pairs in text-rich scenarios, but they are often small in scale and do not include ground-truth text segmentation masks. To address this gap, we introduce a novel data synthesis pipeline that constructs a full-image text image super-resolution dataset. We first apply a text segmentation model, fine-tuned on a bilingual dataset, to large-scale text recognition datasets to extract a diverse set of segmented text patches. These patches are filtered using an OCR model to ensure segmentation accuracy. We then apply a super-resolution model to restore the original quality of these patches and paste them randomly onto high-quality background images drawn from existing general-purpose super-resolution datasets. This process yields our Full-image Text image Super-Resolution (FTSR) dataset, which contains accurate foreground text masks and rich background details.

Our main contributions are summarized as follows:

- We propose TADiSR, a text-aware diffusion-based super-resolution framework that jointly performs image super-resolution and text segmentation. By incorporating a text-aware cross-attention fine-tuning mechanism and joint segmentation decoders, TADiSR effectively enhances text perception and structural fidelity in real-world degraded scenes.

- We propose a scalable data synthesis pipeline for text-aware image super-resolution, enabling the construction of the FTSR dataset with accurate text masks and backgrounds with rich details. The dataset is easily extendable with additional text-oriented images.

- Extensive experiments demonstrate that our approach significantly outperforms prior state-of-the-art models in both qualitative and quantitative evaluations in both synthetic and real-world scenarios, especially in preserving text structure fidelity.

## 2 Related Work

**Real-World Image Super-Resolution.** Real-world image super-resolution (Real-SR) builds upon blind SR by modeling complex, composite degradations that occur in uncontrolled environments. Early methods such as BSRGAN [44] and Real-ESRGAN [32] simulate diverse degradation processes through randomized application orders or multi-stage pipelines, and employ adversarial training to generate visually realistic outputs. Despite their success in producing natural-looking images, GAN-based approaches often fail to accurately recover fine details, especially structural elements like textures and text, due to their limited ability to capture high-level semantics and unstable training dynamics. Recent advances in generative modeling have introduced diffusion-based methods to Real-SR [35, 42, 16], offering improved stability and generative fidelity, yet challenges remain in accurately restoring complex structures, particularly in text-heavy scenarios. ResShift [42] accelerates the denoising sampling of LDM [23] by progressively shifting residuals between LR and HR images during forward propagation. StableSR [29] introduces time-aware encoders, controllable feature wrapping, and novel sampling strategies to fine-tune pre-trained diffusion models, circumventing expensive training costs. SinSR [33] proposes deterministic sampling and consistency-preserving distillation to compress ResShift's sampling into a single-step execution. DiffBIR [16] decomposes blind restoration into degradation removal and detail enhancement stages with adaptive fidelity-generation balance based on regional detail richness. PASD [39] employs ControlNet with degradation-cleaned pixel-domain inputs to enhance pixel-level fidelity. SeeSR [35] develops a degradation-robust tag model generating semantic prompts to improve semantic fidelity in real-world SR. SupIR [41] introduces large-scale high-quality image-text pairs and degradation-robust encoders for latent alignment, complemented by Trimmed ControlNet for efficient restoration control. Wu *et al.* [34] proposed OSEDiff, which directly takes low-resolution images as the starting point for diffusion, and the variational score distillation is applied in the latent space to ensure one-step sampling. Despite notable improvements in realism and fidelity for general content, these methods often overlook or distort text, due to insufficient sensitivity to character-level structures.

**Text Image Super-Resolution.** Text image super-resolution (Text-SR) aims to enhance the legibility of textual content, often by processing cropped image patches containing isolated words or lines. Early approaches, such as those by Dong *et al.* [6], applied general SR techniques like SRCNN [5] to improve OCR performance on low-resolution inputs. TextSR [31] integrates text recognizers into GAN architectures where text recognition loss is backpropagated to guide the generation of legible characters. PlugNet [20] embedded SR units into text recognition training, sharing the backbone to enable more discriminative representations under degraded conditions. TSRN [30] incorporated edge-aware modules and introduced the TextZoom dataset to better simulate real-world text degradation via varying camera focal lengths. Transformer-based methods have further advanced the field. STT [2] leveraged global attention and a dedicated text recognition head to sharpen textual features through position and content-aware losses. TATT [19] proposed a global attention module within CNNs to better handle irregular text layouts. More recent efforts have leveraged powerful generative priors to advance Text-SR: MARCONet [12] combines Transformer backbones with codebooks and StyleGAN priors to recover diverse character styles, while DiffTSR [46] introduced a dual-stream diffusion model that denoises pure text and text image components, respectively, interacting features via a Mixture-of-Modality mechanism. Although these methods achieve state-of-the-art performance on cropped text regions, they struggle to generalize to full images, especially when faced with multi-line, vertical, long, and complex layout text. These methods typically require additional models and complex processing steps to achieve text-aware full-image super-resolution. This highlights the need for a unified and practical solution for full-image text image super-resolution. Furthermore, these works demonstrate the mutual benefits between text image super-resolution and text recognition tasks, suggesting the potential of multi-task learning frameworks. By extending this insight to the full-image solution, our approach integrates text-aware super-resolution and text segmentation into a unified diffusion-based architecture that directly processes whole images.

## 3 Methodology

### 3.1 Overall Architecture

The overall architecture of the proposed TADiSR model is illustrated in Fig. 2. Built upon the Latent Diffusion Model (LDM) framework [23], TADiSR introduces two key components, Text-Aware

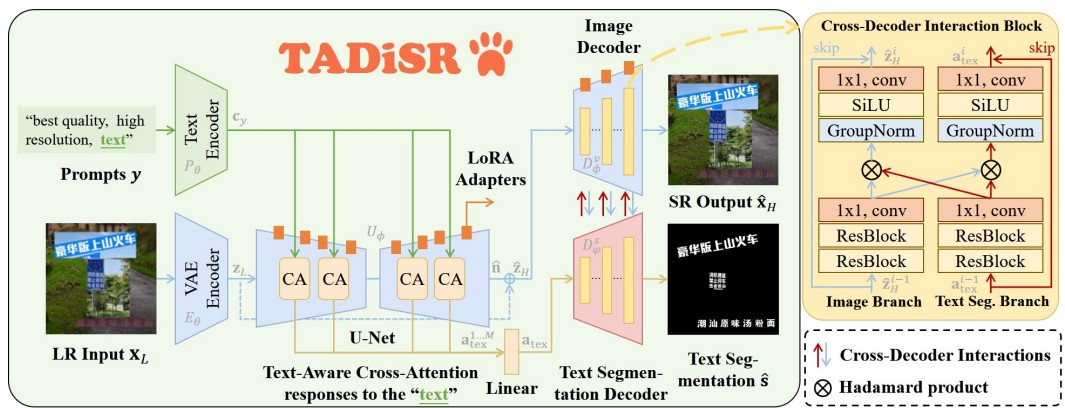

Figure 2: Overall architecture of the proposed TADiSR model. Text-aware cross-attention responses to the word "text" are extracted from the denoising U-Net of the LDM. A Text Segmentation Decoder is introduced, which takes the linearly projected cross-attention response as input and is jointly trained with the original VAE decoder via Cross-Decoder Interaction Blocks (CDIB), forming the Joint Segmentation Decoders (JSD). This design enables simultaneous generation of super-resolved images and text segmentation maps. LoRA adapters are applied to fine-tune the U-Net (including cross-attention layers) and the VAE decoder. The detailed structure of CDIB is depicted on the right.

Cross-Attention (TACA) and Joint Segmentation Decoders (JSD), to equip the diffusion model with fine-grained text perception capabilities.

Given a degraded low-resolution image $\mathbf{x}_L$ and the fixed text prompt $y$, the LR image is encoded by the VAE encoder $E_\theta$ into a latent representation $\mathbf{z}_L = E_\theta(\mathbf{x}_L)$, while the prompt is processed by the text encoder $P_\theta$ to obtain the contextual embedding $\mathbf{c}_y = P_\theta(y)$. We denote the embedding slice corresponding to the keyword "text" as $\mathbf{c}_{\text{tex}}$. Here, $\theta$ represents the parameters of the pre-trained backbone, and $\phi$ denotes the LoRA fine-tuned parameters. During the diffusion process, the latent representation $\mathbf{z}_L$ is denoised by the U-Net backbone to produce the latent $\hat{\mathbf{z}}_H$ by one step:

$$\hat{\mathbf{z}}_H = (\mathbf{z}_L - \beta_t \hat{\mathbf{n}})/\alpha_t, \quad \text{with } \hat{\mathbf{n}} = U_\phi(\mathbf{z}_L; t, \mathbf{c}_y), \tag{1}$$

where $\alpha_t$ and $\beta_t$ are scalars determined by the predefined diffusion time step $t$. The noise estimation $\hat{\mathbf{n}}$ is predicted by the denoising U-Net $U_\phi$. In parallel, the cross-attention maps that respond to $\mathbf{c}_{\text{tex}}$ are linearly projected and fed, along with $\hat{\mathbf{z}}_H$, into the Joint Segmentation Decoders. The decoders yield two predictions: a high-quality super-resolved image $\hat{\mathbf{x}}_H$ and a text segmentation mask $\hat{\mathbf{s}}$. The following two subsections detail the design of the text-aware cross-attention mechanism and the structure of our joint segmentation decoders.

### 3.2 Text-Aware Cross-Attention

In LDM, the cross-attention mechanism [28] is used within the intermediate layers of the U-Net to inject textual conditions into the visual feature stream. It is defined as:

$$\text{CA}(\mathbf{q}, \mathbf{k}, \mathbf{v}) = \text{softmax}\left(\frac{\mathbf{q} \times \mathbf{k}^T}{\sqrt{d}}\right) \times \mathbf{v}, \quad \text{with } \mathbf{q} = \mathbf{W_q} \times \mathbf{z}, \ \mathbf{k} = \mathbf{W_k} \times \mathbf{c}_y, \ \mathbf{v} = \mathbf{W_v} \times \mathbf{c}_y, \tag{2}$$

where $\mathbf{W_q}$, $\mathbf{W_k}$, and $\mathbf{W_v}$ are learnable linear projection matrices, $\times$ denotes matrix multiplication, $\mathbf{z}$ represents intermediate image latent in the U-Net. $\mathbf{q}$, $\mathbf{k}$, and $\mathbf{v}$ are the query, key, and value matrices, respectively, with a dimensionality of $d$.

We enhance the textual content awareness of LDM by guiding the attention responses corresponding to the keyword "text" in the prompt to the text regions. Since the prompt is free-form, the position of the keyword "text" is not fixed. Therefore, we first identify the indices of its token slice $\mathbf{c}_{\text{tex}}$ within the full conditioning vector $\mathbf{c}_y$. These indices are then used to select the corresponding slice from each cross-attention map $\mathbf{a}^m = \mathbf{q}^m \times (\mathbf{k}^m)^T$. The collection of these slices across all $M$ layers is:

$$\mathbf{a}_{\text{tex}}^{1...M} = [\mathbf{a}_{\text{tex}}^1, ..., \mathbf{a}_{\text{tex}}^m, ..., \mathbf{a}_{\text{tex}}^M], \tag{3}$$

where $m \in \{1...M\}$ is the index of a cross-attention layer, and $\mathbf{a}_{\text{tex}}^m$ is the attention slice corresponding to the indices of $\mathbf{c}_{\text{tex}}$. These responses are concatenated along the token channel via Concat($\cdot$) and passed through a linear projection to match the latent dimension of the denoised image code $\hat{\mathbf{z}}_H$:

$$\mathbf{a}_{\text{tex}} = \mathbf{W}_{\mathbf{a}} \times \text{Concat}(\mathbf{a}_{\text{tex}}^{1...M}), \tag{4}$$

where $\mathbf{W}_{\mathbf{a}}$ is the learnable projection matrix. The resulting $\mathbf{a}_{\text{tex}}$ is subsequently fed, together with $\hat{\mathbf{z}}_H$, into the Joint Segmentation Decoders for further joint text segmentation and image super-resolution.

### 3.3 Joint Segmentation Decoders

To enable multi-task learning for both text-aware image super-resolution and text segmentation, we introduce Joint Segmentation Decoders. Specifically, in addition to the original VAE image decoder $D_\phi^v$ in LDM, we design a symmetric Text Segmentation Decoder $D_\varphi^s$. These two decoders jointly decode the denoised image latent $\hat{\mathbf{z}}_H$ and the aggregated text-aware cross-attention $\mathbf{a}_{\text{tex}}$, obtaining $\hat{\mathbf{x}}_H$ (super-resolved image) and $\hat{\mathbf{s}}$ (text segmentation mask) by:

$$\hat{\mathbf{x}}_H = D_\phi^v(\hat{\mathbf{z}}_H), \quad \hat{\mathbf{s}} = D_\varphi^s(\mathbf{a}_{\text{tex}}), \tag{5}$$

where $\phi$ denotes LoRA fine-tuned parameters and $\varphi$ are randomly initialized. Interaction between the two decoders is facilitated via the proposed Cross-Decoder Interaction Block (CDIB), defined as:

$$\hat{\mathbf{z}}_H^i, \mathbf{a}_{\text{tex}}^i = \text{CDIB}^i(\hat{\mathbf{z}}_H^{i-1}, \mathbf{a}_{\text{tex}}^{i-1}), \tag{6}$$

where $i \in \{1, ..., N\}$ indexes the CDIB layers, $\hat{\mathbf{z}}_H^i$ and $\mathbf{a}_{\text{tex}}^i$ denote the image and text segmentation features at the $i$-th layer, respectively.

The structure of the proposed CDIB is illustrated in Fig. 2 (right). CDIB consists of two branches: the Image Branch, which is inserted into the intermediate layers of the VAE image decoder, and the Text Segmentation (TextSeg.) Branch, which is inserted into the corresponding layers of the Text Segmentation Decoder. The input features $\hat{\mathbf{z}}_H^{i-1}$ and $\mathbf{a}_{\text{tex}}^{i-1}$ first pass through two ResBlocks [9]. The resulting features are then processed via a 1x1 convolution and split along the channel dimension. One half is used for within-branch propagation, while the other half is prepared for cross-decoder feature interaction. The exchanged features are passed through a Sigmoid activation and interact with the forward features using Hadamard product, similar in spirit to GLU [24], but applied in a dual-stream setting. The interaction results are then processed by GroupNorm [36], SiLU activation [8], and another 1x1 convolution to produce the final feature maps for each branch. To stabilize training, a skip connection is introduced between the input and output of each branch, with a learnable scaling factor initialized to zero for the residual part.

### 3.4 Loss Function

**SR-oriented Loss.** For the image super-resolution part in our multi-task learning framework, we constrain the reconstruction loss between the predicted high-resolution image $\hat{\mathbf{x}}_H$ and the ground truth high-resolution image $\mathbf{x}_H$ using a weighted sum of MSE, LPIPS, and a modified Focal (MF) loss:

$$\ell_{\text{img}} := \|\hat{\mathbf{x}}_H - \mathbf{x}_H\|_2^2 + \lambda_1 \cdot \text{LPIPS}(\hat{\mathbf{x}}_H, \mathbf{x}_H) + \lambda_2 \cdot \ell_{\text{mf}}, \tag{7}$$

where $\|\cdot\|_2$ denotes the $\ell_2$ norm, and $\lambda_1 = 5.0, \lambda_2 = 10.0$ are balancing coefficients for different loss terms. To enhance the correlation between text structure preservation during image super-resolution and text segmentation, we modify the focal loss [15] by emphasizing hard boundary pixels during image super-resolution guided by segmentation predictions and ground-truths:

$$\ell_{\text{mf}} := \|[\mathbf{1} - \hat{\mathbf{s}} \circ \mathbf{s} - (\mathbf{1} - \hat{\mathbf{s}}) \circ (\mathbf{1} - \mathbf{s})]^\gamma \circ (\nabla \hat{\mathbf{x}}_H - \nabla \mathbf{x}_H)^2\|_1, \tag{8}$$

where $\hat{\mathbf{s}} \circ \mathbf{s} + (\mathbf{1} - \hat{\mathbf{s}}) \circ (\mathbf{1} - \mathbf{s})$ represents the probability of correct classification for each pixel, and $\gamma$ is a hyperparameter adjusting the weight factor. $\circ$ denotes the pixel-wise multiplication. $\|\cdot\|_1$ means the $\ell_1$ norm. $\nabla$ denotes the Sobel edge operator [22]. Edge pixels with high probability of correct classification receive lower weights, whereas those with lower probability receive higher weights, ensuring more attention is given to challenging text structure pixels during image super-resolution. This loss term reinforces the structural accuracy of text regions and strengthens the inter-task relationships in our multi-task learning framework.

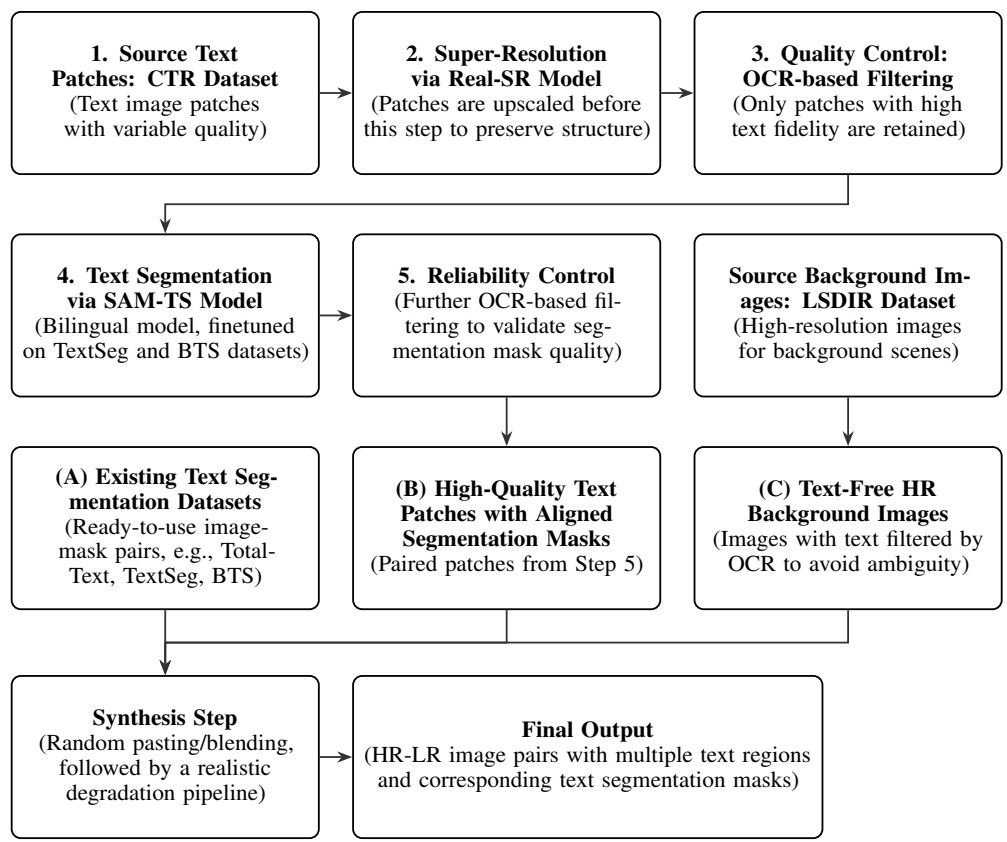

Figure 3: The proposed data synthesis pipeline, which generates training triplets $(\mathbf{x}_L, \mathbf{x}_H, \mathbf{s})$ by blending components from three sources: (a) existing text segmentation datasets; (b) high-quality text patches with aligned masks, produced via an SR-and-filtering pipeline; and (c) text-free HR background images. The synthesized high-resolution images with corresponding segmentation masks are then subjected to a realistic degradation process to generate the final HR-LR-Mask data triplets.

**Segmentation-oriented loss.** For the segmentation prediction $\hat{\mathbf{s}}$ and its ground-truth $\mathbf{s}$, we employ a combination of MSE, Focal, and Dice Losses, which are commonly used in segmentation tasks [40]:

$$\ell_{\text{seg}} := \|\hat{\mathbf{s}} - \mathbf{s}\|_2^2 + \lambda_3 \cdot \text{FocalLoss}(\hat{\mathbf{s}}, \mathbf{s}) + \lambda_4 \cdot \text{DiceLoss}(\hat{\mathbf{s}}, \mathbf{s}). \tag{9}$$

where $\lambda_3 = 10.0, \lambda_4 = 1.0$. The total loss function is formulated as the sum of the above components: $\ell_{\text{tot}} = \ell_{\text{img}} + \ell_{\text{seg}}$, enabling simultaneous learning across both domains for mutual enhancement.

### 3.5 Data Synthesis Pipeline

To enable multi-task learning for text-aware image super-resolution and text segmentation, a dataset consisting of LR images, HR images, and text segmentation maps $(\mathbf{x}_L, \mathbf{x}_H, \mathbf{s})$ is required. While obtaining LR-HR image pairs is easier by applying the Real-ESRGAN [32] degradation pipeline to HR images, acquiring high-quality text segmentation maps is costly in terms of manual annotation. Moreover, existing HR image datasets contain insufficient text-rich samples, whereas OCR datasets with abundant text suffer from inconsistent data quality due to generalization concerns.

As shown in Fig. 3, to solve the dilemma, we propose a data synthesis pipeline that pastes cropped samples from text segmentation datasets onto samples from a high-quality image super-resolution dataset while simultaneously merging the corresponding ground-truth text segmentation maps. Due to the limited availability of fine-grained text segmentation annotations, we train SAM-TS [40] using the TextSeg [37] and BTS [38] datasets to enhance its text segmentation capability. We then apply the trained model to infer text segmentation maps for the CTR [3] dataset and filter reliable pseudo-ground-truth (pseudo-GT) samples by comparing OCR-based text recognition results between

Table 1: Quantitative results on the test set of our proposed FTSR dataset and the validation set of the Real-CE dataset. The best results are displayed in **bold**.

| Datasets | FTSR-TE ($\times 4$) | | | | | Real-CE-val (Aligned, $\times 4$) | | | | |
|---|---|---|---|---|---|---|---|---|---|---|
| Methods | PSNR↑ | SSIM↑ | LPIPS↓ | FID↓ | OCR-A↑ | PSNR↑ | SSIM↑ | LPIPS↓ | FID↓ | OCR-A↑ |
| R-ESRGAN [32] | 22.45 | 0.660 | 0.299 | 65.19 | 0.565 | 21.59 | 0.779 | 0.139 | 44.03 | 0.693 |
| HAT [4] | 24.29 | 0.696 | 0.299 | 66.13 | 0.578 | 23.05 | 0.801 | 0.127 | 44.50 | 0.743 |
| SeeSR [35] | 24.32 | 0.699 | 0.191 | 38.10 | 0.595 | 22.11 | 0.753 | 0.152 | 43.17 | 0.218 |
| SupIR [41] | 22.13 | 0.617 | 0.283 | 48.97 | 0.532 | 21.04 | 0.709 | 0.190 | 46.77 | 0.359 |
| OSEDiff [34] | 24.49 | 0.709 | 0.169 | 32.47 | 0.596 | 21.15 | 0.735 | 0.165 | 50.21 | 0.244 |
| MARCONet [12] | 23.03 | 0.661 | 0.336 | 74.59 | 0.467 | 22.46 | 0.784 | 0.151 | 46.25 | 0.638 |
| DiffTSR [46] | 23.59 | 0.680 | 0.304 | 61.65 | 0.543 | 21.99 | 0.777 | 0.147 | 47.37 | 0.582 |
| **Ours** | **25.49** | **0.736** | **0.152** | **32.13** | **0.662** | **24.02** | **0.829** | **0.100** | **38.01** | **0.882** |

the original images and their segmentation maps. Next, a random number of cropped images are pasted onto high-quality HR samples from the LSDIR [13] dataset, with the corresponding text segmentation maps placed on a zero-initialized image of the same size. To ensure synthesis quality, we filter out CTR samples with a small long-edge/character count ratio and super-resolve them with state-of-the-art Real-SR methods [4]. Additionally, to avoid label ambiguity, we exclude LSDIR samples that contain text by using an OCR-based filtering step. This process results in HR image-text segmentation pairs, which are then degraded by the pipeline of Real-ESRGAN [32] to generate LR images, forming the Full-image Text image Super-Resolution (FTSR) dataset suitable for joint learning of Real-SR and text segmentation.

The primary goal of the synthetic data is to approximate the real-world data distribution and guide the model to capture the core ternary relationship between a low-resolution image $\mathbf{x}_L$, its high-resolution counterpart $\mathbf{x}_H$, and the corresponding text mask $\mathbf{s}$. The model learns a mapping from $\mathbf{x}_L$ to $\mathbf{x}_H$ that must preserve sufficient structural fidelity to also predict $\mathbf{s}$ accurately. From this perspective, while the transition between the pasted text and the background may not always be perfectly seamless, it does not fundamentally alter this core learning objective. In principle, our data synthesis pipeline can be easily extended to generate plentiful new samples. For instance, OCR methods can be used to detect text regions in HR datasets, followed by segmentation using SAM-TS. The segmentation results can then be evaluated with OCR to filter accurate samples, which can be integrated into the synthesis process. The scalable synthetic data reduces the dependency on costly real-world data acquisition and provides a robust foundation for the model, which can be further fine-tuned on smaller, real-world datasets for practical deployment.

## 4 Experiments

### 4.1 Implementation Details

**Dataset Settings.** Our training dataset consists of the proposed synthetic dataset FTSR and the real paired scene text super-resolution dataset Real-CE [18]. The FTSR dataset contains a total of 50,000 triplets $(\mathbf{x}_L, \mathbf{x}_H, \mathbf{s})$, where the first 45,000 triplets are allocated for training, and the remaining 5,000 are used for testing. Since some image pairs in the Real-CE dataset are misaligned [46], we manually filtered out such samples, resulting in 337 training pairs and 189 testing pairs. In this dataset, images captured at a 13mm focal length are used as low-resolution images $\mathbf{x}_L$, while those captured at a 52mm focal length serve as high-resolution images $\mathbf{x}_H$. The text segmentation ground truth is obtained by applying SAM-TS inference on the 52mm images.

**Training Details.** Our model is built upon the Kolors [26] variant of LDMs and follows a tile-based inference strategy similar to SupIR [41]. The diffusion time step $t$ is fixed as 200. We train the model using the PyTorch framework with the AdamW optimizer and a fixed learning rate of $5 \times 10^{-5}$. Training is conducted on four H20 GPUs with a per-GPU batch size of 1 for 200,000 iterations.

### 4.2 Performance Evaluation

As shown in Table 1, we conduct quantitative comparisons between our proposed method and a range of state-of-the-art approaches, including GAN-based (R-ESRGAN [32], HAT [4]), diffusion-based (SeeSR [35], SupIR [41], OSEDiff [34]), and text-specific super-resolution methods (MAR-

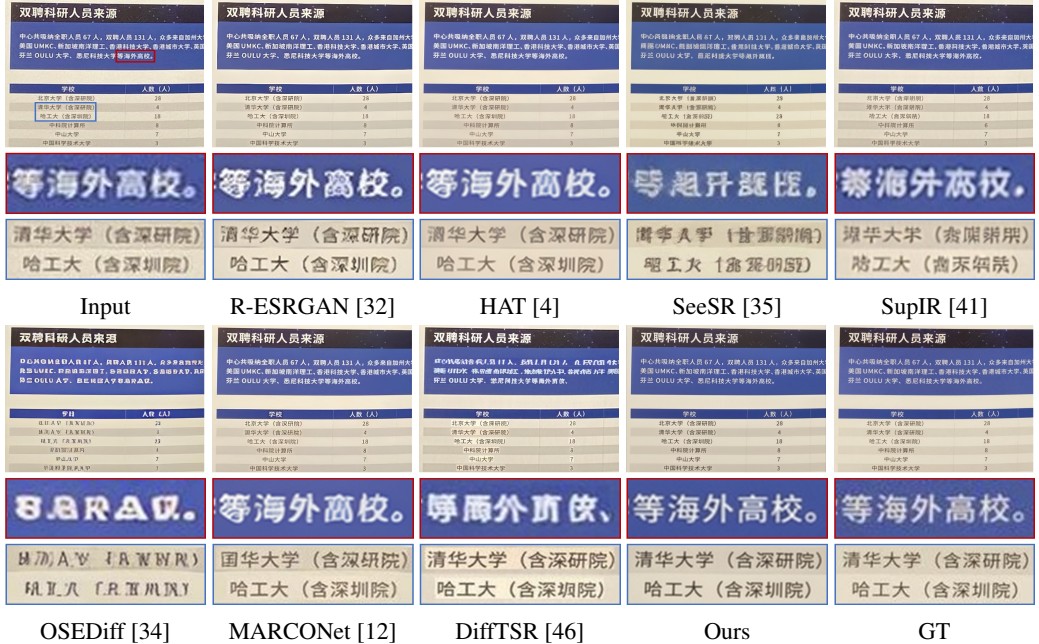

Figure 4: Visual comparison of super-resolution results between previous state-of-the-arts and ours on a sample drawn from the validation dataset of Real-CE [18]. Please note the areas in the boxes.

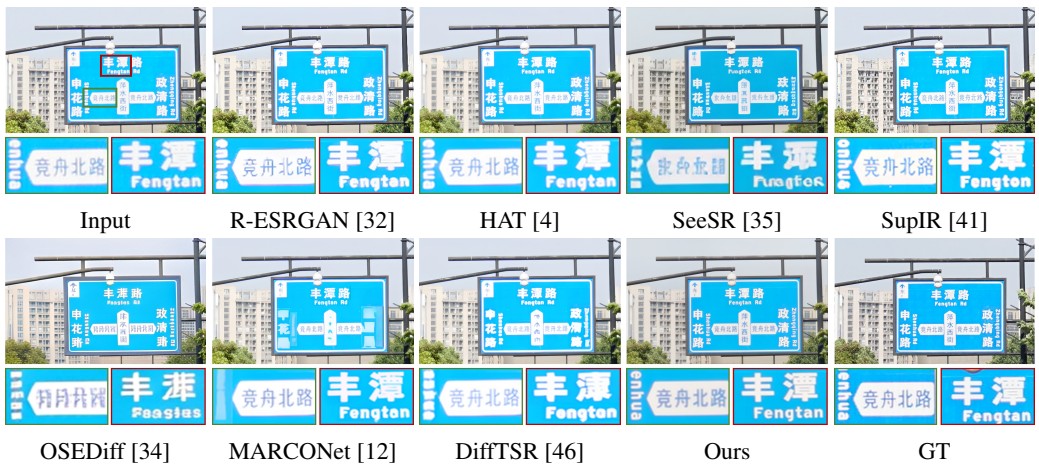

Figure 5: Visual comparison of super-resolution results between previous state-of-the-arts and ours on a sample in real-world scenarios captured in this paper. Please note the areas in the boxes.

CONet [12], DiffTSR [46]). For fair comparison, we fine-tune their released pre-trained models on FTSR and Real-CE using the official training code (if provided). Note that MARCONet and DiffTSR are inherently designed for cropped text region super-resolution and exhibit substantial performance degradation when directly applied to full-image scenarios. To ensure valid outputs for comparison, we first detect and crop text regions using an OCR model [7], then paste their outputs into the super-resolved images produced by HAT [4], which preserves non-text fidelity effectively. We benchmark all methods on two datasets: the test split of our proposed FTSR dataset (FTSR-TE) and the validation set of Real-CE (Real-CE-val), where misaligned LR-HR pairs are manually filtered out. All models are evaluated under a 4× super-resolution setting. We evaluate all methods using PSNR, SSIM, LPIPS [45], FID [10], and OCR-based recognition accuracy (OCR-A), providing a comprehensive assessment across pixel accuracy, perceptual quality, and text restoration quality.

Table 2: Ablation study on different configurations of our proposed design. All models are trained on the FTSR training set and evaluated on the FTSR test set.

| Settings | PSNR↑ | SSIM↑ | LPIPS↓ | FID↓ | OCR-A↑ |
|---|---|---|---|---|---|
| w/o JSD | 25.13 | 0.723 | 0.161 | 33.70 | 0.594 |
| w/o TACA | 25.26 | 0.722 | 0.157 | 32.74 | 0.617 |
| w/o MF Loss | 25.28 | 0.729 | 0.160 | 32.73 | 0.629 |
| Ours | **25.49** | **0.736** | **0.152** | **32.13** | **0.662** |

For the OCR-A metric, we first detect and recognize text in the GT images using an OCR model, preserving the detected bounding boxes. We then extract corresponding regions from the output of each competing method for comparison, focusing on the accuracy of text recognition. The similarity between the prediction and GT is evaluated using the Levenshtein ratio [43], which is defined as $(\text{Len}(r_{\text{pred}}) + \text{Len}(r_{\text{gt}}) - \text{Dist}(r_{\text{pred}}, r_{\text{gt}}))/(\text{Len}(r_{\text{pred}}) + \text{Len}(r_{\text{gt}}))$, where $\text{Len}(\cdot)$ extracts the string length, $r_{\text{pred}}$ denotes the recognition result of a competing method, $r_{\text{gt}}$ represents the recognition result in the GT image, and $\text{Dist}(\cdot, \cdot)$ refers to the Levenshtein distance between two strings.

As can be seen, our proposed TADiSR demonstrates overall superiority across pixel-level, perceptual, and text accuracy metrics on both synthetic and real-world datasets. Notably, on the Real-CE dataset, TADiSR outperforms the second-best method (HAT) by a large margin of **18.7%** in OCR-based recognition accuracy (OCR-A), highlighting its exceptional capacity in recovering accurate textual content from degraded inputs.

We further conduct visual comparisons on the Real-CE dataset, as shown in Fig. 4. Due to real-world degradations, the input images often exhibit irregular noise, blurred strokes, and stroke adhesions. GAN-based methods, while effective at denoising, fail to recover incorrect text structures, leaving adhesion artifacts. Diffusion-based generic SR methods, though possessing stronger generative capabilities, lack text-aware guidance and thus tend to produce more severe stroke distortions, sometimes resulting in completely unrecognizable patterns. Methods tailored for cropped text regions (e.g., MARCONet, DiffTSR) show better structure restoration for short texts. However, MARCONet enforces a hard limit of 16 tokens per text block. As a result, longer regions are neglected and actually reconstructed using HAT (in the red box). Moreover, when OCR guidance is inaccurate, these OCR-dependent methods may even generate semantically incorrect characters. DiffTSR, though not subject to hard length constraints, performs reliably only for texts shorter than 8 characters, and also fails to produce valid results for longer sequences. Both MARCONet and DiffTSR rely on box-wise inference and post-processing fusion, often introducing visible artifacts at patch boundaries during full-image reconstruction. TADiSR effectively avoids the aforementioned pitfalls, making it more suitable for practical applications.

To further evaluate generalization, we collected additional real-world samples using a digital camera. As illustrated in Fig. 5, TADiSR delivers comparable super-resolution quality on non-text regions while significantly enhancing blurred and adhesive strokes. Moreover, it is worth noting that our method effectively avoids the severe performance degradation commonly observed in text image super-resolution models when handling vertically arranged text. Addressing this issue in existing methods typically requires additional vertical training samples and longer training durations, whereas our approach naturally circumvents this limitation. Due to space constraints, we include more visual results under various bilingual scenarios in the supplementary material for interested readers.

## 4.3 Ablation Study

To verify the effectiveness of our design, we conducted an ablation study, as shown in Table 2 and Fig. 6. Specifically, we evaluate three ablated variants: (1) removing the Joint Segmentation Decoders (w/o JSD), (2) disabling the Text-Aware Cross-Attention mechanism (w/o TACA), and (3) eliminating the modified Focal Loss introduced in Sec. 3.4 (w/o MF Loss). All variants exhibit performance degradation to varying degrees, with the w/o JSD model suffering the most significant drop, an 11.4% decrease in OCR accuracy, alongside declines in pixel fidelity and perceptual metrics. From the qualitative results in Fig. 6, we can also observe that removing any of the proposed modules weakens the model's ability to reconstruct accurate text structures. The absence of Joint Segmentation Decoders deprives the model of explicit structural supervision, making it learn text shapes in a less

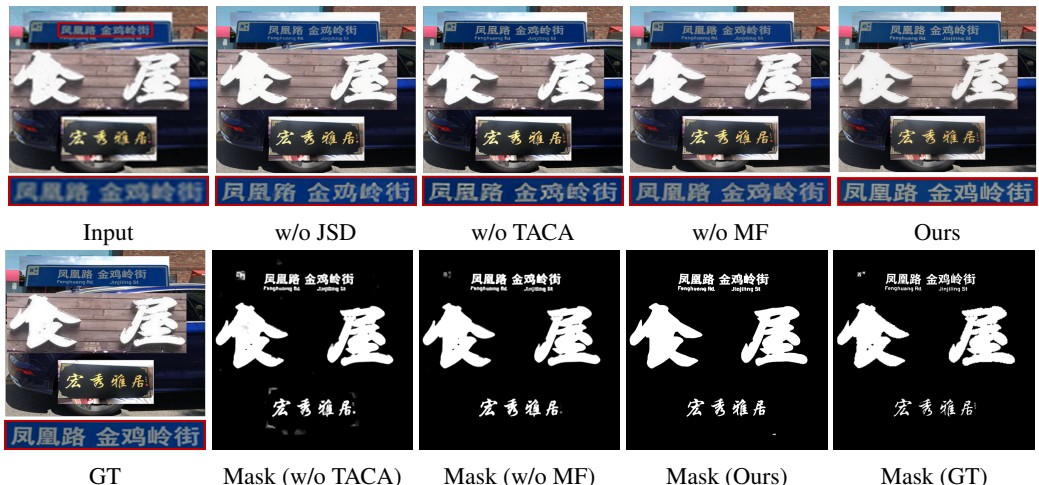

Figure 6: Visual results of the ablation study under different configurations. Note the zoom-in regions, where our full model achieves more accurate text super-resolution with clearer structural fidelity compared to its ablated variants. Predicted text segmentation masks are also provided for reference.

guided and inefficient manner. Without Text-Aware Cross-Attention, the predicted segmentation masks erroneously highlight irrelevant non-text regions, which mislead the model during training and impair its structural reconstruction ability. The modified Focal Loss, which serves to couple the segmentation and super-resolution outputs, encourages the model to focus more on structurally ambiguous regions around text strokes. Eliminating this loss leads to noticeably blurrier and less well-defined text structures, which also hinders the accuracy of the text segmentation predictions.

Overall, the ablation results confirm that each component in our framework directly supports better text structure restoration. Removing any of them leads to noticeable drops in both visual fidelity and quantitative performance, confirming the practical effectiveness of our design choices.

## 5 Conclusion

In this paper, we proposed TADiSR, a text-aware diffusion model designed for real-world image super-resolution with joint text segmentation decoding. By finetuning the cross-attention layers within the LDM, our approach encourages the model to pay more attention to text-rich regions. Furthermore, the introduction of cross-decoder interactions allows structural text information to be shared between the super-resolution and segmentation decoding branches, resulting in high-quality image outputs with accurate and well-preserved text structures. Extensive experiments on both synthetic and real-world datasets demonstrate the overall superiority of our method in terms of both quantitative metrics and visual fidelity. There are also several intriguing directions that merit further investigation, such as extending cross-attention tuning strategies to other rare semantic categories to improve the LDM's generation accuracy, and developing dedicated diffusion-based text segmentation frameworks. Additionally, while TADiSR shows superior performance, there remains significant room for improvement. For instance, collecting more real-world paired data across varied focal lengths, conducting more precise text segmentation annotations, or incorporating stronger priors that explicitly link pure text representations to stylized text appearances could further enhance its performance and generalization in real-world scenarios. Overall, our method offers a feasible and efficient solution for full-image text-aware super-resolution, and we hope it draws greater attention from the community to this practically important but underexplored area.

## Acknowledgement

We would like to thank Jinwei Chen in vivo for the insightful discussions and feedback. This work was partially supported by the National Natural Science Foundation of China under Grant no. 62372251.

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

## A    FTSR Synthetic Dataset Visualization

To further illustrate the construction and diversity of our proposed dataset FTSR, we present several example triplets $(\mathbf{x}_L, \mathbf{x}_H, \mathbf{s})$ in Fig. 7. Our proposed synthesis pipeline allows for free composition between background images and foreground text regions. Moreover, we leverage existing text segmentation dataset samples to further train a text segmentation model, which is then used to infer on large-scale text recognition datasets. We thereby extract a substantial number of text region images with accurate segmentation masks using OCR-based filtering method. As a result, the samples feature rich and varied background contexts, diverse font styles, and accurate pixel-level text segmentation masks, offering supervision for joint learning of super-resolution and segmentation tasks.

## B    Text Segmentation Comparison

In Fig. 8, we provide a qualitative comparison between our TADiSR model and a state-of-the-art segmentation method Hi-SAM on real-world degraded samples collected in this paper by a digital camera. Despite that part of our training segmentation labels were selected from Hi-SAM outputs, our model still delivers significantly finer and more precise segmentation results than those of Hi-SAM across varied scenes. This highlights the strong mutual benefits between text-aware super-resolution and text segmentation tasks, and further validates the effectiveness of our joint learning strategy.

## C    Extended Visual Comparisons on Real-World Samples

We provide additional visual comparisons between our method and previous state-of-the-art approaches in Figs. 9–12, using real-world samples collected in this paper in the wild. To comprehensively assess the generalization capabilities of each method, the examples span diverse and challenging scenarios including overexposed neon signs at night, long horizontal text, vertically engraved characters, and handwritten text. In Fig. 9, overexposure leads to severe character merging in neon signs, a situation where existing methods struggle to recover legible text. In contrast, our method successfully reconstructs distinct character structures, also showing superior performance on small-scale text. Fig. 10 illustrates results on long horizontal text. While GAN-based models largely preserve structure but fail to enhance quality, diffusion-based generic SR methods introduce structural distortions due to lack of text awareness. MARCONet [12] fails to generate valid outputs on long text due to its hard length constraint, with most content falling back to HAT outputs. Similarly, DiffTSR [46] struggles with long sequences, and even in valid predictions (green boxes), character sticking occurs. Our model, however, improves visual quality while maintaining accurate character shapes throughout. In Fig. 11, which contains vertically engraved characters, both GAN and generic diffusion methods suffer from reduced legibility and structural distortions. Text image SR methods generally fail due to the vertical layout. Our method, benefiting from global text-awareness, produces sharp and coherent results, enhancing faint strokes. For handwritten text in Fig. 12, other methods result in missing or merged strokes, while our approach accurately reconstructs the undermined handwritten structure. Additionally, we present more results on the Real-CE dataset in Figs. 13 and 14, which focus on text-rich scenarios like poster and book covers. Our model consistently restores fused strokes and outputs structurally accurate super-resolved results, even for fine-grained characters.

## D    Limitation Analysis

As illustrated in Fig. 15, we showcase a challenging example from the Real-CE dataset. In our result, the character highlighted by the red circle exhibits a structural discrepancy when compared with the ground truth. This particular character has a high stroke density, and in the input image, the strokes are heavily fused due to real-world degradation, making it difficult to visually discern the original structure—even for a human observer. Consequently, all methods, including text image super-resolution approaches (within their limitation of text length), fail to predict the correct structure of this character. Future work could explore interactive strategies such as text mask editing or user-guided correction to address such hard cases. Nonetheless, it is worth emphasizing that apart from this one character, our method consistently outperforms previous approaches across the rest of the bilingual text in the image, producing clearer and more accurate structural reconstructions.

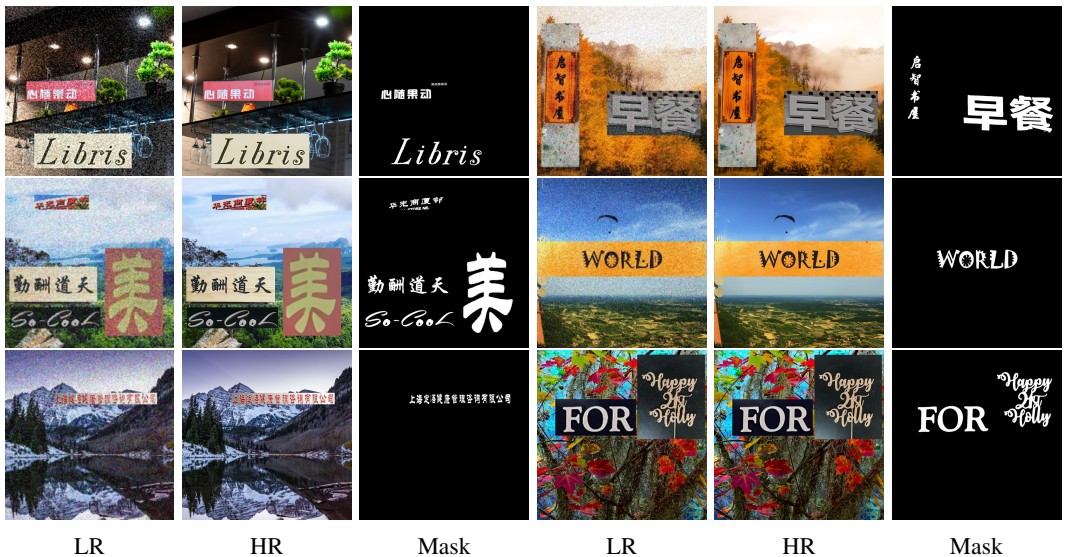

| LR | HR | Mask | LR | HR | Mask |

Figure 7: Sample triplets from our FTSR dataset, including low-resolution inputs (LR), high-resolution references (HR), and ground truth segmentation masks (Mask).

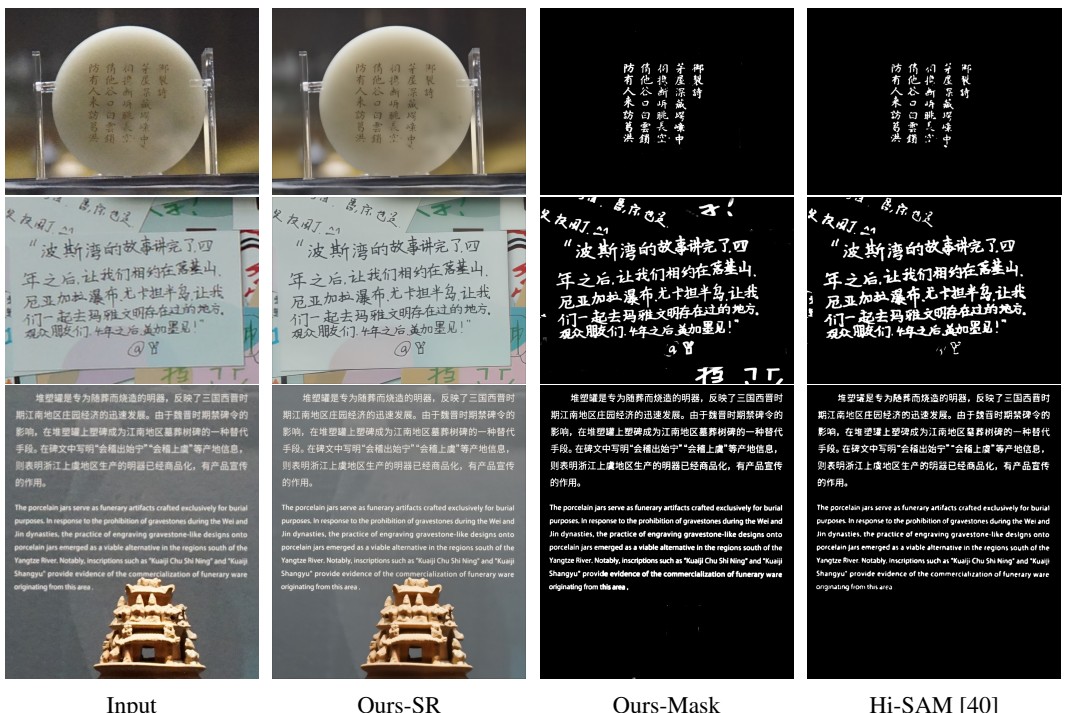

| Input | Ours-SR | Ours-Mask | Hi-SAM [40] |

Figure 8: Text segmentation comparison between TADiSR and Hi-SAM [40] on real-world degraded cases, including carved, handwritten, and long printed text. Our results are noticeably finer and more accurate than those of the dedicated text segmentation model Hi-SAM.

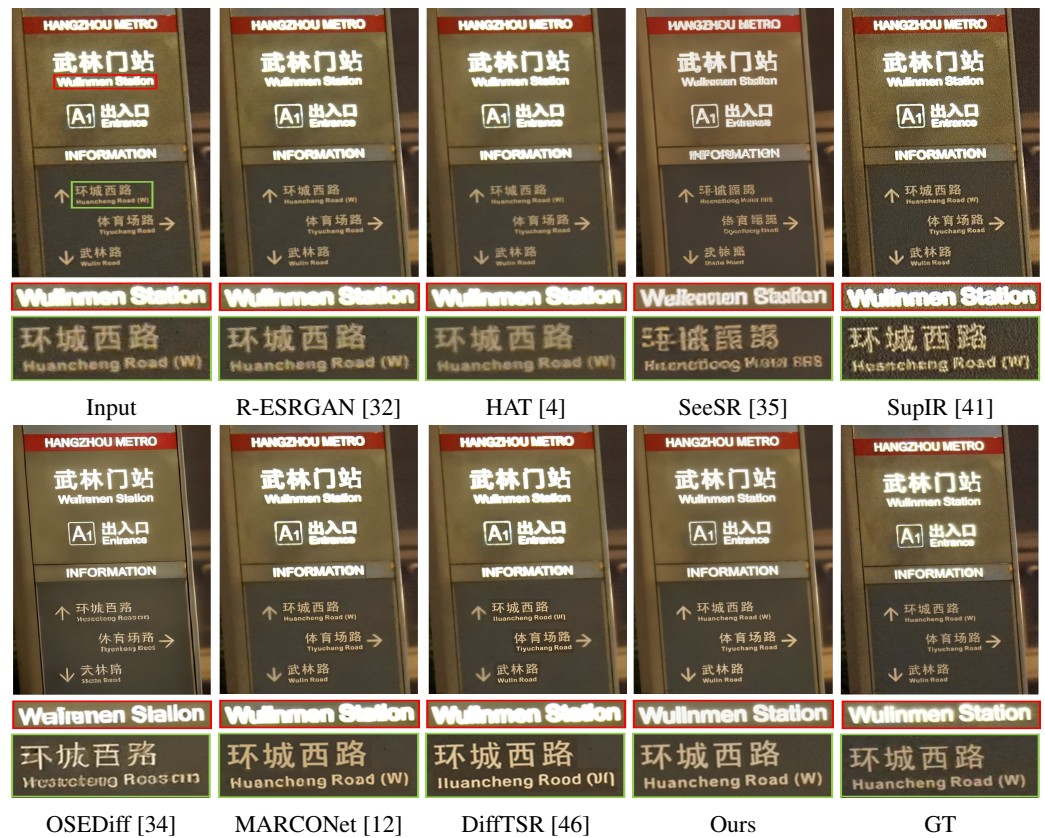

Figure 9: Visual comparison of super-resolution results between previous state-of-the-arts and ours on a sample in real-world scenarios captured in this paper. Please note the areas in the boxes.

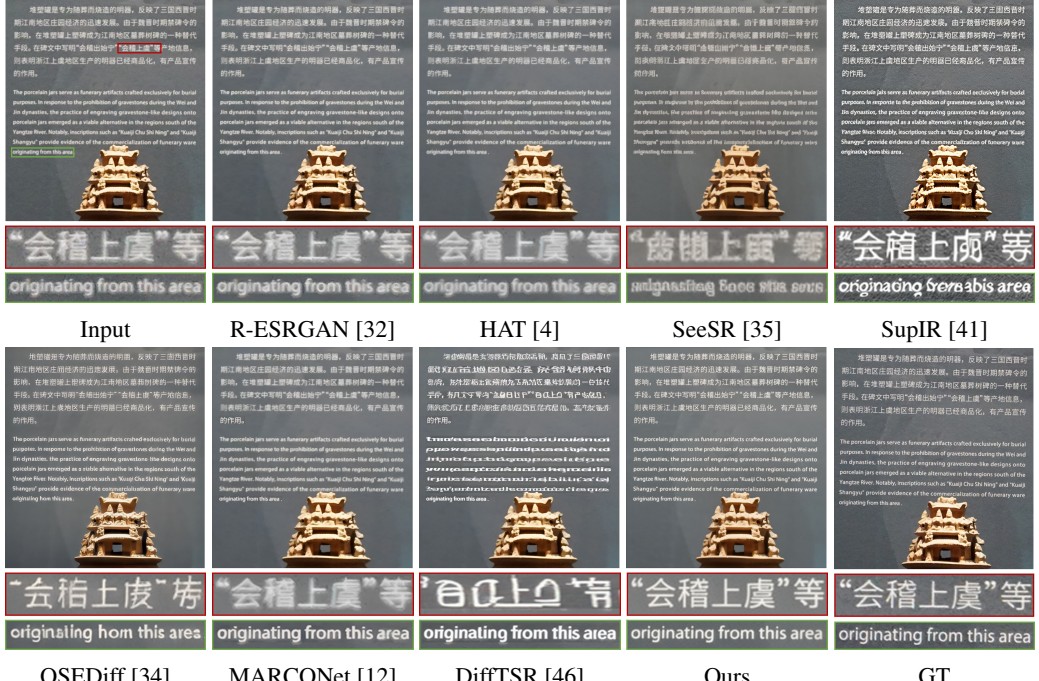

Figure 10: Visual comparison of super-resolution results between previous state-of-the-arts and ours on a sample in real-world scenarios captured in this paper. Please note the areas in the boxes.

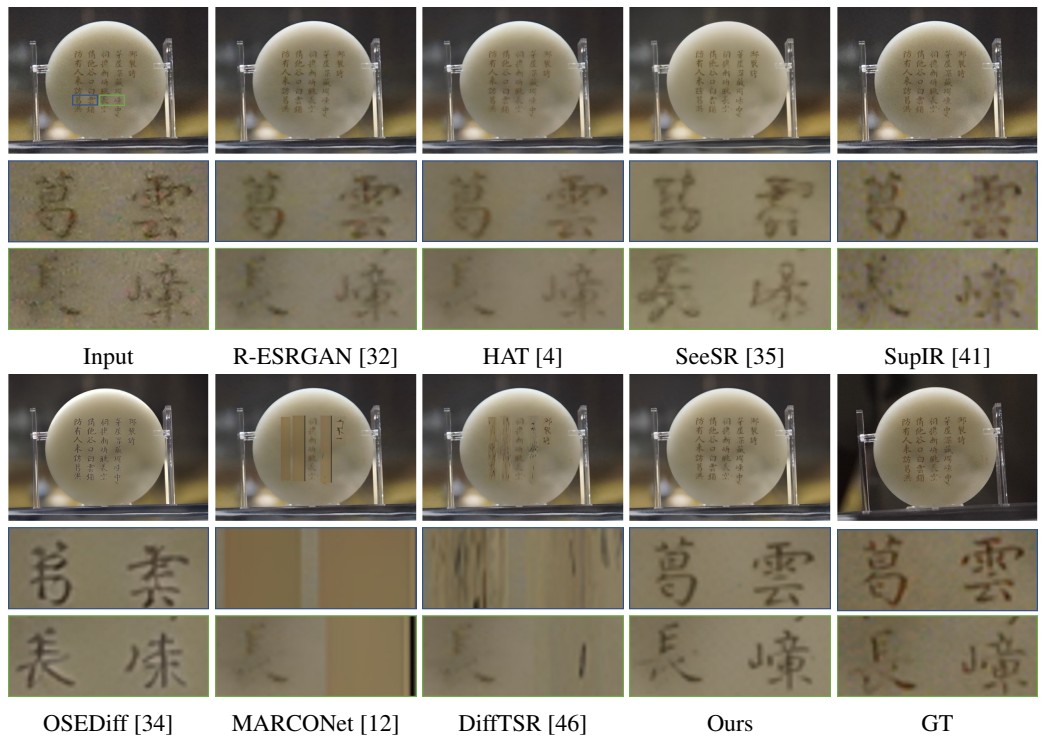

Figure 11: Visual comparison of super-resolution results between previous state-of-the-arts and ours on a sample in real-world scenarios captured in this paper. Please note the areas in the boxes.

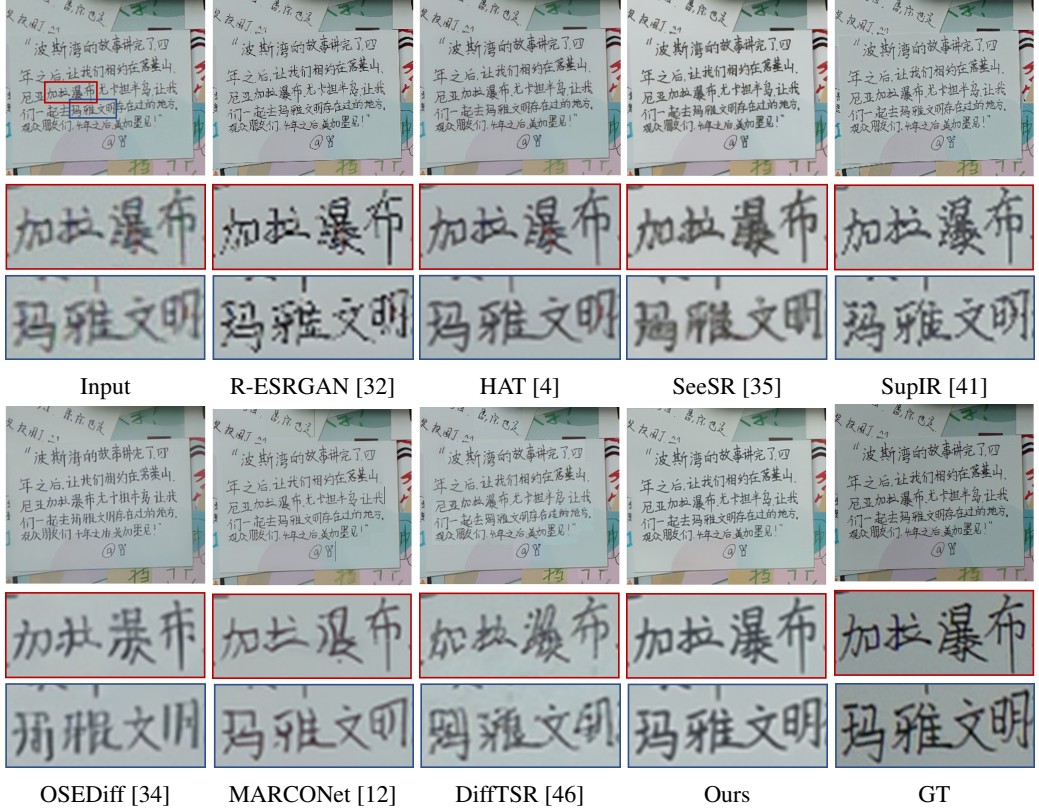

Figure 12: Visual comparison of super-resolution results between previous state-of-the-arts and ours on a sample in real-world scenarios captured in this paper. Please note the areas in the boxes.

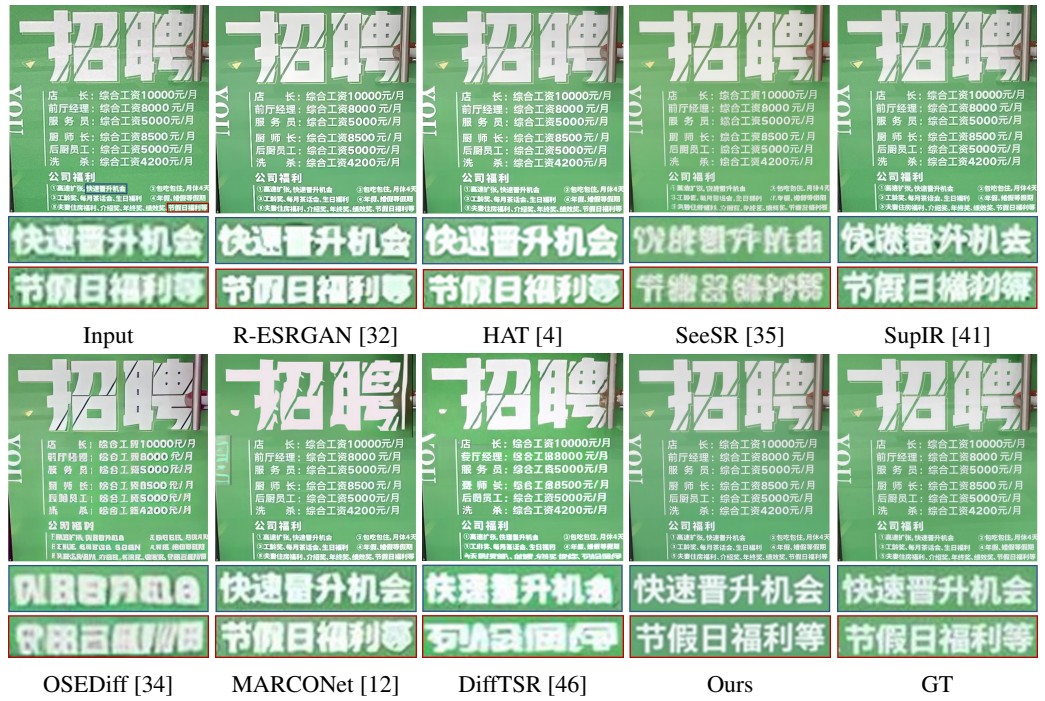

Figure 13: Visual comparison of super-resolution results between previous state-of-the-arts and ours on a sample drawn from the validation dataset of Real-CE [18]. Please note the areas in the boxes.

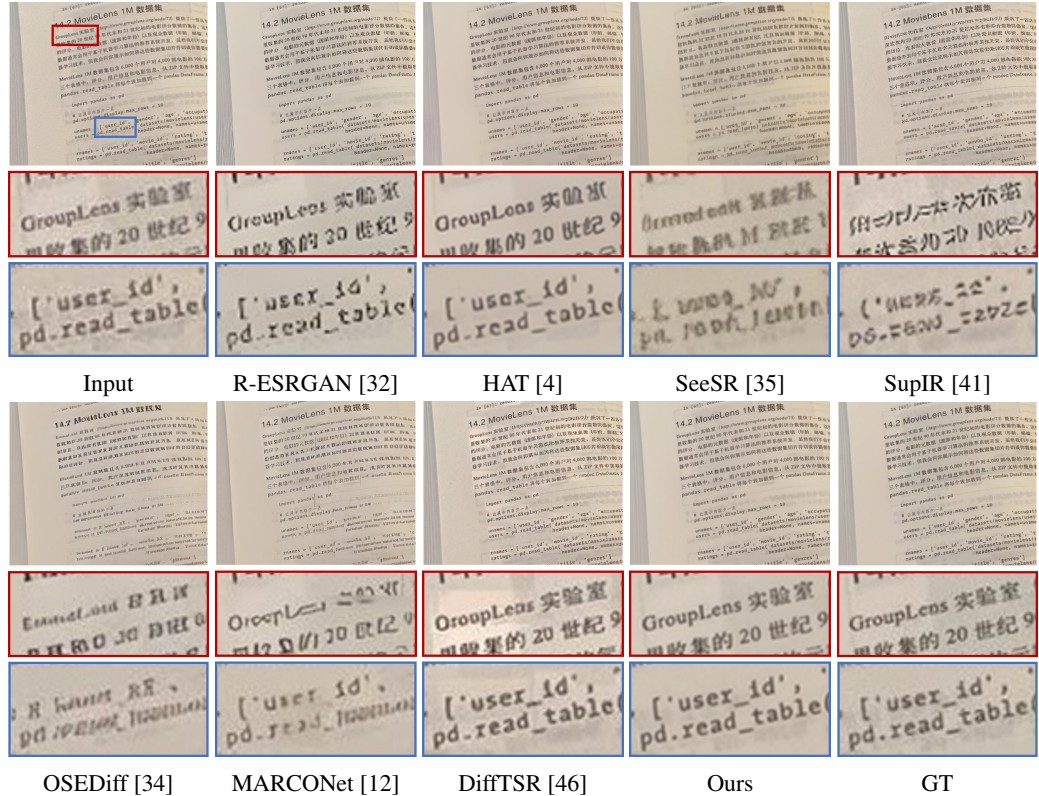

Figure 14: Visual comparison of super-resolution results between previous state-of-the-arts and ours on a sample drawn from the validation dataset of Real-CE [18]. Please note the areas in the boxes.

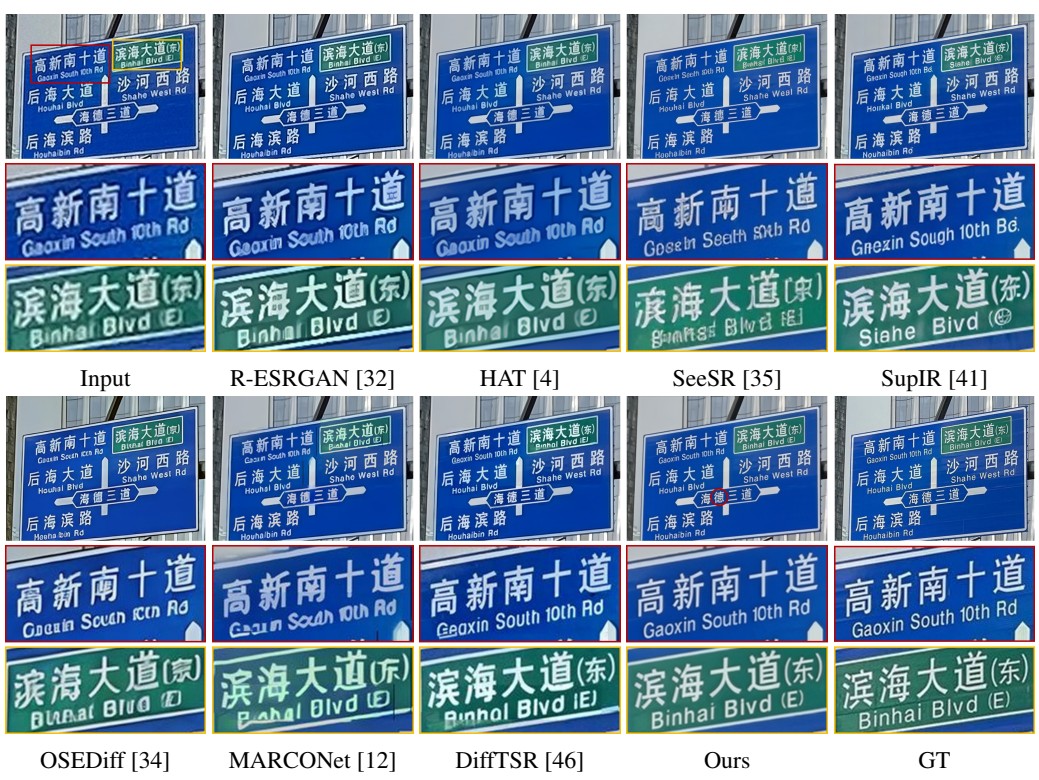

Figure 15: A case from the Real-CE dataset showing our limitation. The character highlighted by the red circle shows structure deviation due to extreme stroke fusion, while other characters are well reconstructed by our method.

