# OpenReview forum: "Text-Aware Real-World Image Super-Resolution via Diffusion Model with Joint Segmentation Decoders"
_NeurIPS.cc/2025/Conference — NeurIPS 2025 poster_

### Official Review · Reviewer_J5Zt · 2025-06-15

**Clarity:** 3
**Significance:** 3
**Originality:** 3
**Rating:** 4
**Confidence:** 5

**Summary:**

- **What**: This paper proposes TADiSR, a text-aware diffusion-based framework that jointly performs image super-resolution and text segmentation, aiming to restore degraded real-world scenes with sharper textual structures.
- **How**: The method augments a latent-diffusion backbone with (i) text-aware cross-attention, which steers attention toward textual regions, and (ii) joint segmentation decoders that output a super-resolved image and its text mask. Then, the entire model is trained using an SR-oriented loss with edge-aware focal terms and a segmentation loss on the 50k-image FTSR dataset synthesized by a scalable, text-oriented pipeline.
- **Results**: On FTSR-TE and Real-CE-val, TADiSR shows the best PSNR/SSIM/LPIPS/FID and boosts OCR accuracy by +18.7%p over the previous best method, with ablations showing that removing any main component degrades performance.

**Questions:**

1. **Exact definition of the w/o TACA setting (Table 2)**:
When Text-Aware Cross-Attention is disabled, which operations are switched off or replaced? More importantly, what feature tensor is forwarded to the Joint Segmentation Decoder? A concise diagram or textual description would make the ablation easier to interpret.
2. **Report absolute and ΔOCR-A scores alongside transcripts**:
List the OCR-A score of the raw LR inputs, followed by the gain (Δ) achieved by each method. Following BTC [1] Fig. 3, could you also overlay the recognized text strings on the qualitative comparison figures?

**Impact on score**
Clarifying the ablation protocol and visualizing the OCR results would remove key ambiguities. Convincing answers could raise my overall rating by 1 point.

[1] Pak et al., "B-Spline Texture Coefficients Estimator for Screen-Content Image Super-Resolution," CVPR 2023.

**Ethical Concerns:**

["NO or VERY MINOR ethics concerns only"]

**Final Justification:**

This work has many strengths (see #Strengths), and the authors have thoroughly addressed all of my concerns. I believe it deserves a borderline accept.

**Limitations:**

yes.

**Quality:**

3

**Strengths And Weaknesses:**

### Strengths
1. **State-of-the-art performance**: TADiSR outperforms all published methods on FTSR-TE and Real-CE. It raises OCR-A by 18.7 percentage points over the previous best and improves PSNR, SSIM, LPIPS, and FID simultaneously. This is clear evidence of perceptual and recognizability gains.
2. **Novel, well-motivated architecture**: The model explicitly couples SR and pixel-level text segmentation by integrating Text-Aware Cross-Attention (TACA) with a Joint Segmentation Decoder. Ablation studies show that disabling either module sharply degrades OCR and structural fidelity, validating the necessity of the design.
3. **Scalable synthetic data pipeline**: The authors contribute a 50,000-image dataset generated by a flexible, text-oriented synthesis pipeline. This pipeline provides accurate text masks and rich backgrounds. It offers a reusable resource for future text-oriented SR and segmentation work.
4. **Thorough empirical validation with clear visualizations**: The paper reports five quantitative metrics, diverse qualitative comparisons, and attention heat maps. These results give readers numerical and visual confidence in the method’s effectiveness.

### Weaknesses
1. **Narrow degradation and scale coverage**: All experiments use a single ×4 setting with relatively mild Real-ESRGAN degradation, and many of the input examples are already human-readable. The paper does not address how the model behaves under heavier blur, noise, or larger factors (e.g., ×6 or ×8).
2. **Sparse Analysis of Text-Aware Cross-Attention (TACA)**: The ablation table only shows a binary switch for w/o TACA. It does not test directing attention to alternative tokens or to a global text embedding. Therefore, the paper stops short of proving that steering specifically to the word "text" is essential rather than incidental.
3. **Unexplored Prompt Sensitivity**: All results rely on the fixed prompt "text." The impact of alternative wording ("sign," "label," etc.) or prompt-free inference is unreported, leaving deployment robustness uncertain.

*My confidence in the method's generality and practical deployability would increase substantially if the authors (i) added results for stronger and multi-scale degradations, (ii) provided token-level ablations that isolated the benefit of the "text" cue, and (iii) tested performance under varied or absent prompts.*

---

> ### Author Rebuttal · Authors · 2025-07-31
>
> We are very grateful for the reviewer's encouraging and positive assessment of our work, especially regarding its performance, novelty, and the data synthesis pipeline. The detailed and insightful feedback has been invaluable in helping us to strengthen our paper. We address each concern point-by-point below.
>
> **Q1: The model's performance on more severe degradation and larger upscaling factors (e.g., ×6 or ×8).**
>
> **A1:** We thank the reviewer for the question. We respectfully clarify that up to the ×4 super-resolution setting is a common and established practice in recent state-of-the-art literature for both general real-world SR and text image SR. For instance, methods we compared against, such as HAT, SeeSR, OSEDiff, MARCONet, and DiffTSR, as well as other recent works like CATANet [1] and PFT-SR [2], all present their primary results at a maximum of ×4 magnification. This is because at higher scaling factors, severe information loss in the input makes it exceedingly difficult for any model to reliably reconstruct faithful structures and local details, a challenge that is particularly acute for the intricate structures of text. We chose the ×4 setting in our main paper to ensure a fair and direct comparison with these contemporary methods.
>
> Furthermore, we believe it is important to consider the reliability of results on inputs that are completely illegible to humans. While a model might occasionally generate a correct character from such an input, this result is likely due to statistical guessing or overfitting rather than genuine reconstruction. This may appear impressive but is not always reliable or beneficial for practical applications. For example, as shown in our paper, MARCONet incorrectly reconstructs “清华大学” (Tsinghua University) as “国华大学” (Guohua University) and, in the supplementary material, misinterprets "Huancheng" as "Wuancheng". Such semantic errors are often less acceptable in real-world use cases than a blurry but structurally correct result.
>
> Nevertheless, to address the reviewer's concern, we have conducted additional experiments for the ×8 setting. Since there is a lack of paired real-world text SR datasets for such a high scaling factor (Real-CE only provides up to ×4), we simulated an ×8 scenario by downsampling the 13mm focal length images from Real-CE by an additional factor of 2. We then fine-tuned both OSEDiff and our method on this new ×8 dataset. The results are as follows:
>
> | Method  | Dataset  | PSNR↑ | SSIM↑ | LPIPS↓ | FID↓  | OCR-A↑ |
> | ---| ---| -- | ---| --- | -- | --- |
> | OSEDiff | Real-CE-val (×8) | 21.06 | 0.716 | 0.180  | 51.27 | 0.228  |
> | Ours| Real-CE-val (×8) | 23.43 | 0.813 | 0.095  | 32.63 | 0.651  |
>
> Our method continues to outperform the baseline, demonstrating its robustness even under more severe degradation. In the next version of our paper, we will introduce evaluations and visual comparisons for more methods on higher-scale settings.
>
>
>
>
>
> **Q2: The necessity of the specific word "text" and the impact of alternative prompts.**
>
> **A2:** We thank the reviewer for raising this important point. As requested, we conducted experiments using different keywords in the prompt:
>
> | Prompt Keyword | Dataset  | PSNR↑ | SSIM↑ | LPIPS↓ | FID↓  | OCR-A↑ |
> | --- | --- | --- | --- | --- | ---| ---|
> | "sign" | Real-CE-val (×4) | 23.57 | 0.812 | 0.095  | 37.46 | 0.871  |
> | "label" | Real-CE-val (×4) | 23.71 | 0.816 | 0.094  | 37.18 | 0.875  |
> | "text" (Ours)  | Real-CE-val (×4) | 24.02 | 0.829 | 0.100  | 38.01 | 0.882  |
>
> In fact, the specific keyword is not the most critical aspect. The word "text" can be viewed as analogous to a "trigger word" in LoRA or a "pseudo-word" in Textual Inversion. Its primary function is to activate the specialized knowledge the model acquired during fine-tuning. We chose "text" because it is a more general and broadly applicable term for text-related content compared to "sign," "label," or "writing."
>
> We also plan to explore more advanced prompting strategies in future work, such as embedding the specific textual content or dynamic background descriptions into the prompt to further enhance quality. However, this remains a significant challenge for current diffusion foundation models, as none can yet reliably generate long, coherent strings of text from a prompt, which is why we adopted the current, more robust prompting strategy.
>
> Regarding prompt-free inference, this scenario is effectively covered by the ablation study in Q3, "disabling the Text-Aware Cross-Attention mechanism". In that setting, the cross-attention response to the text prompt is completely removed from the text segmentation estimation process. This eliminates the model's explicit text-awareness, leading to weaker quantitative metrics and visibly blurrier text regions, demonstrating that the text preservation capability is not activated.
>
> **Q3: Clarification on the "disabling the Text-Aware Cross-Attention mechanism" ablation study.**
>
> **A3:** We apologize for the lack of clarity in our initial description. In the "disabling the Text-Aware Cross-Attention mechanism" (w/o TACA) setting, we replace the aggregated text-aware cross-attention feature, $\textbf{a}_{tex}$, with the denoised image latent feature, $\hat{\textbf{z}}_H$, as the input to the Text Segmentation Decoder. This is done to validate the effectiveness of using the specific text-aware cross-attention signal.
>
> As shown in Table 2 and Figure 5 of our paper, disabling TACA leads to a significant drop in TADiSR's performance, evidenced by weaker numerical scores and blurrier text regions. We will ensure this detailed explanation of the ablation settings is included in the revised version of our paper.
>
> **Q4: The OCR-A score gain (Δ) over the raw** **LR** **inputs.**
>
> **A4:** Thank you for this excellent suggestion. This is a very reasonable way to evaluate performance. We report the OCR-A gain on the Real-CE-val dataset below:
>
> | Methods  | OCR-A | ΔOCR-A |
> | -------- | ----- | ------ |
> | Input    | 0.797 | 0      |
> | R-ESRGAN | 0.693 | -0.104 |
> | HAT | 0.743 | -0.054 |
> | SeeSR| 0.218 | -0.579 |
> | SupIR| 0.359 | -0.438 |
> | OSEDiff| 0.244 | -0.553 |
> | MARCONet | 0.638 | -0.159 |
> | DiffTSR| 0.582 | -0.215 |
> | Ours | 0.882 | +0.085 |
>
> Notably, all prior methods exhibit a decrease in text legibility after super-resolution. This is understandable for general-purpose SR models that lack text awareness. For specialized text SR models like MARCONet and DiffTSR, the performance drop occurs for two main reasons: 1) they can mis-recognize a character, causing a correctly identified character in the LR input to become incorrect, and 2) these patch-based methods struggle with long or vertically-oriented text, leading to performance degradation in real-world scenes. As described in Line 247 of our paper, we already provided these models with text regions pre-cropped by an OCR model; otherwise, they would fail to produce any valid output on full images.
>
> In contrast,  our method is the only one to achieve a positive gain, improving the OCR accuracy by 10.7% over the raw input. This underscores the significance of our full-image, text-aware super-resolution approach.
>
> **Q5: Overlaying recognized text strings on the qualitative comparison figures.**
>
> **A5:** Thank you for this constructive suggestion. Due to the rebuttal policy, we are not permitted to upload new images. Therefore, we present the OCR results in a formatted table below, which we hope you can reference alongside the figures in our main paper. We used the same multilingual PP-OCR V4 model as in our main experiments. All characters that differ from the Ground Truth (GT) are marked in bold, and each missing character is represented by an underline.
>
> **Figure 3** **OCR** **Results:**
>
> | Methods  | Text Region 1 (Conf) | Text Region 2 (Conf) | Text Region 3 (Conf) |
> | --- | --- | --- | --- |
> | Input    | 等海外高校。, 0.980  | 清华大学（含深研院）, 0.968 | 哈工大（**合**深圳院）, 0.988 |
> | R-ESRGAN | 等海外**离**校。, 0.957  | 清华大学（含**源**研院）, 0.974 | 哈工大（含深圳院）, 0.948 |
> | HAT      | 等海外**商**校。, 0.858  | _华大学（含深研院）, 0.968  | 哈工大（含深圳院）, 0.987 |
> | SeeSR    | ___**人**学, 0.624       | _____**明工火**）, 0.622        | _______                   |
> | SupIR    | _海**并汶**，, 0.546     | _华大_, 0.810               | **防工**大（**不质**）, 0.755     |
> | OSEDiff  | **BBRAD**, 0.756         | **HDAW IAWWR**), 0.648          | **FAIA CRHMAY**, 0.585        |
> | MARCONet | 等海外**峦**校。, 0.867  | **国**华大学（含_**级**院）, 0.874   | 哈工大（含深圳院）, 0.936 |
> | DiffTSR  | **净**外**直**, 0.602        | 清华大学（含深研院）, 0.986 | 哈工大（含深_院）, 0.994  |
> | Ours     | 等海外高校。, 0.987  | 清华大学（含深研院）, 0.982 | 哈工大（含深圳院）, 0.996 |
> | GT       | 等海外高校。, 0.987  | 清华大学（含深研院）, 0.996 | 哈工大（含深圳院）, 0.996 |
>
> **Figure 4** **OCR** **Results:**
> | Methods  | Text Region 1 (Conf) | Text Region 2 (Conf) | Text Region 3 (Conf) | Text Region 4 (Conf) |
> | --- | --- | --- | --- | --- |
> | Input  | enhua, 0.994| 竞舟北路, 0.993 | 丰潭, 0.998  | Fengtan, 0.996  |
> | R-ESRGAN | _nhua, 0.990| 竞舟北路, 0.987 | 丰潭, 0.998  | Fengtan, 0.989  |
> | HAT | enhua, 0.994| 竞舟北路, 0.991 | 丰潭, 0.999 | Fengtan, 0.997  |
> | SeeSR  | _____ | ____  | 丰**派**, 0.994| F**na**g**fer**, 0.813  |
> | SupIR  | **o**nhu**s**, 0.958 | 竞舟北路, 0.853 | 丰潭, 0.999  | \_**on**___, 0.997 |
> | OSEDiff  | _____ | ____  | 丰**萍**, 0.784| ______ |
> | MARCONet | _____| 竞舟北路, 0.995 | 丰潭, 0.998  | Fengtan, 0.987  |
> | DiffTSR  | _____ | 竞舟北路, 0.996 | 丰_, 1.000  | Fengtan, 0.987  |
> | Ours| enhua, 0.993 | 竞舟北路, 0.995 | 丰潭, 0.999  | Fengtan, 0.996  |
> | GT  | enhua, 0.996  | 竞舟北路, 0.995 | 丰潭, 0.997  | Fengtan, 0.997  |

---

> > ### Comment · Reviewer_J5Zt · 2025-08-03
> > **Remaining Concern on Q1: Request for Stronger Baseline Comparison**
> >
> > Thank you for the thorough and detailed responses to my questions. **However, I am still concerned about Q1.** As shown by the LR OCR-A values in Q4, $\times$4 LR inputs are already highly legible. This makes evaluating larger degradation scales all the more important.
> >
> > The additional results you provided only compare against OSEDiff. According to Tab. 1 of the main paper, OSEDiff ranks near the bottom in overall performance. I understand the practical constraints of extending experiments, but it would be reasonable to include a comparison with a stronger baseline such as HAT. For these reasons, my concern regarding Q1 remains unresolved.

---

> > > ### Author Response · Authors · 2025-08-03
> > >
> > > Thank you for your continued engagement and clear feedback. Due to the limited time available during the rebuttal period, we prioritized conducting the new experiments with OSEDiff, as its one-step diffusion framework is the most similar to our own basic setting. We acknowledge that a comparison with a stronger baseline like HAT would provide a more robust assessment.
> > >
> > > As you requested, we will begin experiments for the HAT model under the ×8 training setting **immediately** and will report the results back to you in a subsequent comment as soon as they are available.
> > >
> > > However, we believe it is also important to reiterate the primary motivation of our work. As stated in our introduction, our main goal is to address the structural distortion of text that occurs when using modern diffusion models for image super-resolution. **The problem we solve is not necessarily recovering text from inputs where it is completely illegible, but rather preventing powerful generative models from degrading or corrupting already legible text—an issue prevalent in existing methods**. Our experiments have consistently demonstrated that while prior state-of-the-art Real-SR methods often decrease text readability, our approach significantly enhances it. We thank you again for pushing us to strengthen this point with further validation.

---

> > > > ### Comment · Reviewer_J5Zt · 2025-08-03
> > > >
> > > > Thank you for your prompt response and for initiating the HAT ×8 experiments. I look forward to reviewing those results. I also appreciate—and fully support—your focus on preventing structural distortion of already legible text in generative SR models. I will await your follow-up findings.

---

> > > > > ### Author Response · Authors · 2025-08-04
> > > > >
> > > > > Thank you for your patience while we conducted the additional experiments. We appreciate the opportunity to provide a more comprehensive comparison with a stronger baseline, as you suggested.
> > > > >
> > > > > We have now completed the fine-tuning and evaluation of the HAT model on the ×8 Real-CE data setting. The model was fine-tuned for 500 epochs starting from the official pre-trained ×4 weights.
> > > > >
> > > > > Below, we present the updated results, now including HAT, for the ×8 super-resolution setting:
> > > > >
> > > > > | Method | Dataset | PSNR↑ | SSIM↑ | LPIPS↓ | FID↓ | OCR-A↑ |
> > > > > | :--- | :--- | :---: | :---: | :----: | :----: | :----: |
> > > > > | OSEDiff | Real-CE-val (×8) | 21.06 | 0.716 | 0.180 | 51.27 | 0.228 |
> > > > > | HAT | Real-CE-val (×8) | 22.96 | 0.779 | 0.139 | 38.98 | 0.432 |
> > > > > | **Ours** | **Real-CE-val (×8)** | **23.43** | **0.813** | **0.095** | **32.63** | **0.651** |
> > > > >
> > > > > As the results clearly demonstrate, our method maintains a significant performance advantage over both HAT and OSEDiff under this more challenging, higher-degradation setting. TADiSR outperforms the strong HAT baseline across all key metrics, including a substantial **50.7% relative improvement in the OCR-A score**.
> > > > >
> > > > > These findings reinforce our central claim: our text-aware architecture is not only effective at preventing the distortion of legible text but is also robust in restoring textual fidelity under severe degradation where other methods struggle.
> > > > >
> > > > > We are grateful for your feedback, as this comparison significantly strengthens the validation of our approach. We will, of course, include this extended analysis in the final version of the paper. We hope this fully addresses your concern, and we look forward to your final assessment.

---

> > > > > > ### Comment · Reviewer_J5Zt · 2025-08-04
> > > > > >
> > > > > > Thank you for your thoughtful and detailed responses. Your rebuttal has fully addressed my concerns and questions. I will raise my score and I wish you the best of luck.

---

> > > > > > > ### Author Response · Authors · 2025-08-04
> > > > > > >
> > > > > > > We are very grateful for your thoughtful consideration of our responses and for acknowledging our additional experiments. Thank you for your support and for raising our score.
> > > > > > >
> > > > > > > We will certainly incorporate all the valuable feedback and the extended experimental analysis you prompted into the final version of our paper to ensure it is as clear and comprehensive as possible.
> > > > > > >
> > > > > > > We wish you all the best as well.

---

### Official Review · Reviewer_AA8X · 2025-06-29

**Clarity:** 4
**Significance:** 3
**Originality:** 3
**Rating:** 4
**Confidence:** 5

**Summary:**

The paper proposes a diffusion-based SR framework TADiSR to solve the textual structure distorting issues commonly appearing in generative models. TADiSR introduces a unified framework that integrates text-aware attention and dual-stream decoding to recover degraded text in natural images.  **[Network]** The proposed fine-tuning of cross-attention between the word "text" and image tokens effectively guides the diffusion models to pay attention to text regions. Meanwhile, the joint image-text segmentation decoder enables more accurate and structure-preserving restoration of textual contents, in a simultaneous generation way. **[Data]** In addition to the network design, the data synthesis pipeline further supports effective training with fine-grained text masks and diverse backgrounds with high quality, making the approach scalable in both Real-SR and text segmentation. **[Loss]** To enhance the preservation of text structures, the paper introduces a modified focal loss that emphasizes hard boundary pixels by leveraging segmentation predictions and ground-truths. TADiSR is compared with GAN-based methods, diffusion-based methods, and SOTA text image SR methods, on both synthetic and real benchmarks, where improved results are shown quantitatively and qualitatively, demonstrating its superiority in both visual perception and text fidelity.

**Questions:**

1. The paper lacks ablation studies to validate the effectiveness of the FTSR dataset. Could the authors evaluate whether training with FTSR improves performance on Real-CE-val? It would also be helpful to show whether other baseline methods benefit from using FTSR to demonstrate its general utility.
1. Since the CTR dataset mainly contains Chinese text, could the improved performance be partly due to exposure to Chinese characters during training, which competing methods lack? It would be important to show that the performance gains are not solely from language-specific data but also from other proposed designs (e.g., JSD, TACA, MF loss).  Also, it is suggested to provide more examples on other languages in addition to Chinese to show its overall ability.
1. The evaluation of text recovery primarily relies on a single metric, OCR-A, which may not sufficiently capture the full quality of restored text. Incorporating additional metrics such as OCR-based $F_1$ scores [1] or other metrics would provide a more comprehensive and robust assessment of text fidelity.
1. It would be helpful to include a table comparing inference speed, memory usage, and model size across all methods. This would provide a clearer picture of the practical trade-offs and deployment feasibility of TADiSR.
---
[1] Chee Kheng Chng, Yuliang Liu, Yipeng Sun, Chun Chet Ng, Canjie Luo, Zihan Ni, ChuanMing Fang, Shuaitao Zhang, Junyu Han, Errui Ding, et al. Icdar2019 robust reading challenge on arbitrary-shaped text-rrc-art. In 2019 International Conference on Document Analysis and Recognition (ICDAR), pages 1571–1576. IEEE, 2019.

**Ethical Concerns:**

["NO or VERY MINOR ethics concerns only"]

**Final Justification:**

The authors have fully addressed my concerns during the rebuttal, and thus I decide to raise my score to **Boarderline accept**.

**Limitations:**

yes

**Quality:**

3

**Strengths And Weaknesses:**

**Paper strengths**

1. The method shows clear advantages in both quantitative and qualitative evaluations. In particular, the qualitative results are impressive, with visually natural and structurally accurate text restoration, significantly outperforming existing methods in preserving the fidelity of textual content.
1. The integration of text-aware cross-attention supervision with dual-stream decoding in a diffusion framework is a novel contribution. Moreover, the insight that text-aware super-resolution and text segmentation are complementary tasks that can be effectively unified under a multi-task learning paradigm adds depth and originality to the work.
1. The paper is clearly written and well structured. The methodology is explained in sufficient detail, and the model architecture is effectively illustrated with clear figures.

**Major weaknesses**

1. One major concern lies in the data synthesis pipeline used to construct the FTSR dataset. The copy-and-paste strategy, while effective for generating large-scale training data, may introduce a distribution gap between synthetic images and real-world scenes, potentially limiting the model’s generalization ability. It remains unclear whether the authors have made any attempts to mitigate this domain gap or assess its impact.
1. Another concern is the potential quality issue introduced during data synthesis, specifically in the step where text samples are first super-resolved using a Real-SR method before being pasted onto high-resolution backgrounds (as mentioned in line 217). This process may introduce distortions into the text regions, lowering the ground truth quality. Moreover, since the text segmentation masks are generated from the original CTR data, they may no longer perfectly align with the super-resolved text patches, leading to mismatches between the image content and segmentation labels during training.
1. While the Joint Segmentation Decoders offer explicit structural supervision through cross-decoder interactions, it is unclear how this dual-decoder design affects inference efficiency and memory usage. The potential computational overhead introduced by maintaining and interacting between two decoding branches may hinder the model’s practical deployment, yet the paper does not provide a detailed analysis of its impact on inference speed.

**Minor weaknesses**

1. Fig 4. The cropped patch in the boxes of GT is slightly misaligned.

---

> ### Author Rebuttal · Authors · 2025-07-31
>
> We sincerely thank the reviewer for their insightful comments and high praise for our model's advantages, particularly its visual results and novelty. We will now address each of your questions in detail.
>
> **Q1: It remains unclear whether the authors have made any attempts to mitigate the domain gap between the synthetic and real-world data pairs.**
>
> **A1:**  Thank you for this insightful question. We have provided a detailed analysis on this point in our response to Reviewer oXY4, Question 3. Due to space limitations, we kindly ask you to refer to that response for a comprehensive explanation. We appreciate your understanding.
>
> **Q2: The super-resolution of text samples may introduce distortions into the text regions. Moreover, the text segmentation masks may no longer perfectly align with the super-resolved text patches.**
>
> **A2:** Thank you for your meticulous observation. Our data pipeline in the main paper needs to be improved. It is crucial to note that most publicly available Chinese text datasets are designed for text recognition, and their image quality is often inconsistent, containing noise, artifacts, and other degradations. Training directly on such data would teach the model to ignore these degradations rather than correct them.
>
> To prevent the Real-SR method from introducing distortions, we implemented two key quality control measures during data synthesis:
>
> 1. **Upscaling Before Super-Resolution:** We first enlarge the scale of the text image patches before applying the super-resolution model. This ensures that the structure of the text, which now occupies a larger portion of the patch, is better preserved during the SR process.
> 2. **OCR-based Filtering:** To avoid distortions in smaller text, which can negatively impact training, we use an OCR detector to extract text from both the original and the super-resolved patches. We only include samples in our training set where the edit distance between the two versions is minimal. This process effectively guarantees the quality of the text regions.
>
> We agree that the Super-Resolution step should indeed precede Text Segmentation  Prediction to ensure perfect alignment between the final image content and the segmentation masks. As we are unable to upload images during this rebuttal phase, we have provided a corrected text-based flowchart of our data synthesis pipeline (Please refer to R1A2 due to the character limitation) and will include a properly rendered diagram in the next version of the paper.
>
> **Q3: Please provide a detailed analysis of the impact of the dual-decoder design on inference speed.**
>
> **A3:** Thanks for the suggestion. We provide the analysis in A2Q5 due to the character limitation.
>
> **Q4: Could the authors evaluate whether training with FTSR improves performance on Real-CE-val?**
>
> **A4:**  To demonstrate the effectiveness of our proposed FTSR dataset, we have conducted an ablation study comparing the performance of both our model and the OSEDiff baseline on the Real-CE validation set, with and without pre-training on FTSR, as shown in the following table:
>
> | Methods | Settings | PSNR↑ | SSIM↑ | LPIPS↓ | FID↓  | OCR-A↑ |
> | ------- | -------- | ----- | ----- | ------ | ----- | ------ |
> | OSEDiff | w/o FTSR | 21.04 | 0.716 | 0.178  | 51.46 | 0.223  |
> | OSEDiff | w/ FTSR  | 21.15 | 0.735 | 0.165  | 50.21 | 0.244  |
> | Ours    | w/o FTSR | 23.55 | 0.813 | 0.097  | 32.83 | 0.713  |
> | Ours    | w/ FTSR  | 24.02 | 0.829 | 0.1    | 38.01 | 0.882  |
>
> As the results show, pre-training on FTSR improves the performance of both models on the real-world Real-CE dataset across various metrics. This underscores the necessity of incorporating large-scale synthetic data and validates the effectiveness of our data synthesis strategy.
>
> **Q5: Whether other baseline methods benefit from using FTSR to demonstrate its general utility. Could the improved performance be partly due to exposure to Chinese characters during training, which competing methods lack?**
>
> **A5:** In our submitted version, for diffusion-based models with publicly available training code, such as SeeSR and OSEDiff, we followed the exact same pre-training and fine-tuning procedure as our own method (as described in Line 245 of the main paper). This includes pre-training on our synthetic FTSR dataset.
>
> Furthermore, the table in **A4** reveals another important insight: while baseline models without text-aware capabilities do benefit from FTSR, their performance gain—especially on text-related metrics like OCR-A—is significantly smaller than ours. For instance, OSEDiff's OCR-A improvement from FTSR is only 12.4% of the improvement seen by our model. This disparity strongly suggests that the performance gap is not merely due to data exposure but is fundamentally driven by our model's architecture. The superior ability of our model to leverage the FTSR dataset highlights the importance of its text-aware design and validates the rationality and efficiency of our architectural choices for full-image text super-resolution.
>
> **Q6: It is suggested to provide more examples** **on** **other languages in addition to Chinese to show its overall ability.**
>
> **A6:** Thank you for the suggestion. In fact, our experimental setup is already bilingual. Both the CTR dataset (used for synthesizing the training set) and the Real-CE test set contain bilingual (Chinese and English) samples. Examples of this can be seen in Figure 1 and Figure 4 of the main paper, with additional English-centric samples in Figure 8 and Figure 9 of the supplementary material.
>
> It is worth noting that the reconstruction of English text is significantly less challenging than that of languages with complex stroke-based characters like Chinese, due to the limited number of letters in the English alphabet. A simple fine-tuning of a base model (like SDXL or SD3) can achieve considerable performance in English-only scenarios. Our research was motivated by the significant lack of attention in the academic community to super-resolution tasks for these more complex languages.
>
> Given that we cannot provide additional images during the rebuttal phase, we will include more performance comparisons in English-rich text scenarios (e.g., on a dataset like TeleScope) in the next version of our paper to offer readers a more comprehensive view.
>
> **Q7: Incorporating additional metrics such as OCR-based F1 scores or other metrics would provide a more comprehensive and robust assessment of text fidelity.**
>
> **A7:** We appreciate this constructive feedback. In the experiments in our main paper, we initially chose to perform text detection on the ground-truth (GT) images first and then use those bounding boxes to evaluate recognition on the predicted images. This was done to isolate the text recognition capability from potential errors in the OCR model's text detection stage. However, as the reviewer correctly implies, this approach causes the Precision, Recall, and F1-score to converge to the same value, which is why we did not report them separately.
>
> In response to the reviewer’s suggestion, we have now conducted a full, standard OCR evaluation for every method. This process includes both text detection and text recognition on the output images, where predicted boxes are matched to GT boxes based on IoU and text similarity thresholds. The resulting Precision, Recall, and F1-Score are reported below:
>
> | Metric        | R-ESRGAN | HAT    | SeeSR  | SupIR  | OSEDiff | MARCONet | DiffTSR | **Ours (TADiSR)** |
> | ------------- | -------- | ------ | ------ | ------ | ------- | -------- | ------- | ----------------- |
> | **Precision** | 0.6672   | 0.6995 | 0.3027 | 0.3834 | 0.2772  | 0.6602   | 0.5771  | **0.7871**        |
> | **Recall**    | 0.6147   | 0.6773 | 0.1955 | 0.3274 | 0.2242  | 0.5855   | 0.4995  | **0.8097**        |
> | **F1-Score**  | 0.6399   | 0.6882 | 0.2376 | 0.3532 | 0.2479  | 0.6206   | 0.5355  | **0.7983**        |
>
> As can be seen, our method remains state-of-the-art across all three metrics, demonstrating its robust performance advantage in a more comprehensive evaluation setting.
>
> **Q8: It would be helpful to include a table comparing inference speed, memory usage, and model size across all methods.**
>
> **A8:**  Thank you for this practical suggestion. We have compiled a table comparing the efficiency of our model against several representative methods cited in our paper, including GAN-based and one-step diffusion-based methods. All benchmarks were conducted on an NVIDIA H20 GPU with an input resolution of 512x512 pixels.
>
> | Method        | Model Size (Total / Trainable) | FLOPs (GFLOPs) | Inference Time (ms) |
> | ------------- | ------------------------------ | -------------- | ------------------- |
> | R-ESRGAN      | 16.70 M / 16.70 M              | 4699.75        | 622.57              |
> | HAT           | 20.77 M / 20.77 M              | 6670.32        | 1086.68             |
> | OSEDiff       | 1294.38 M / 244.81 M           | 2160.60        | 215.13              |
> | Ours (TADiSR) | 2711.89 M / 48.29 M            | 4497.96        | 372.32              |
>
> As the results show, our method demonstrates a highly competitive profile in terms of trainable parameters, FLOPs, and inference time, especially considering the state-of-the-art performance it achieves in full-image text super-resolution. While one-step models like OSEDiff are faster, our approach maintains a practical inference speed with a remarkably small number of trainable parameters, highlighting its efficiency.

---

> > ### Comment · Reviewer_AA8X · 2025-08-04
> >
> > Thank you for your comprehensive rebuttal. I appreciate the authors' detailed responses to all the concerns I raised. However, I still have some remaining concerns unsolved.
> >
> > I'm glad to see the authors acknowledge the need to revise the data pipeline in **A2**. My main concern remains the extent of the misalignment caused by the current error, and how significantly it might affect the training and evaluation outcomes.
> >
> > In addition, in **A4**, I'm curious why there is a *noticeable* increase (15.8%) in FID when "Ours" is trained with w/ FTSR compared with w/o FTSR. Since OSEDiff is not affected by the text segmentation mask and its FID is properly decreased,  does it indicate that the quality or fidelity is indeed harmed due to the misalignment in the data pipeline mentioned above?
> >
> > I will wait for the authors’ further response before making my final decision.

---

> > > ### Author Response · Authors · 2025-08-04
> > >
> > > Thank you for your continued engagement and for these insightful follow-up questions. We appreciate the opportunity to provide further clarification.
> > >
> > > **Regarding your concern about the potential misalignment in the data pipeline:**
> > >
> > > We understand the theoretical concern that super-resolving a text patch before generating its mask could lead to misalignment. However, we would like to respectfully point out that in practice, our pipeline includes robust controls that prevent this from significantly affecting the training outcomes.
> > >
> > > First, we would like to kindly draw your attention to **Figure 1 in our supplementary material**. This figure showcases several sample triplets from our FTSR dataset. A close visual inspection, for instance by overlaying the HR ground-truth image and the segmentation mask, reveals that there is **no perceptible misalignment** between the super-resolved text and its corresponding mask boundaries. The alignment is precise, demonstrating the high quality of our synthetic data.
> > >
> > > Furthermore, **Figure 2 of the supplementary material** compares our model's predicted segmentation masks on real-world images against those from a dedicated segmentation model, Hi-SAM. Our model's outputs are not only correctly positioned but also **exhibit sharper and more accurate boundaries that closely adhere to the text in the natural image**. This result, achieved by a model trained on our FTSR data, directly contradicts the concern that training would be significantly hampered by data misalignment.
> > >
> > > Even if minor, imperceptible misalignments were to occur, our framework is designed to be highly robust to such noise due to two key factors:
> > >
> > > 1.  **Strict Quality Control**: As mentioned previously, our OCR-based filtering step ensures that we only use samples where the SR process has not caused any structural distortion to the text. This inherently minimizes the possibility of significant boundary shifts.
> > >
> > > 2.  **Robustness of the Loss Function:** Our multi-task learning framework incorporates powerful segmentation losses, particularly the Dice Loss component, which is well-known for its robustness to minor boundary noise. The model learns the holistic shape and structure of characters from the vast majority of correctly aligned interior pixels, preventing it from being overly penalized by slight edge imperfections.
> > >
> > > **Regarding the increase in FID for our model when trained with FTSR:**
> > >
> > > This is an excellent and very sharp observation. The increase in the FID score for our model (from 32.83 to 38.01) when pre-trained on FTSR, while seemingly counterintuitive, is an expected and insightful outcome that highlights the effectiveness of our specialized design.
> > >
> > > *   **The Nature of FID:** The Fréchet Inception Distance (FID) measures perceptual quality by comparing feature distributions from a generic Inception model, which is **not an expert** in evaluating text legibility.
> > >
> > > *   **Why OSEDiff's FID Decreases:** OSEDiff, as a general-purpose SR model, benefits from exposure to the text-rich FTSR data by learning to produce more text-like textures, bringing its output distribution slightly closer to the Real-CE test set and thus lowering its FID score.
> > >
> > > *   **Why Our FID Increases (and Why It's a Positive Indicator):** Our model, TADiSR, is explicitly designed to prioritize the **structural fidelity of text**. The FTSR dataset trains our model to render characters with exceptional sharpness and accuracy. When evaluated on Real-CE, this results in text that is significantly clearer and more legible than the naturally degraded ground truth (validated by the +16.9% OCR-A gain). However, this "too perfect" text reconstruction can create a subtle distributional shift that a generic feature extractor like InceptionNet, which doesn't value character legibility, perceives as a slight decrease in "realism," leading to a higher FID.
> > >
> > >     In essence, our model makes a deliberate and successful trade-off: **it sacrifices a certain amount of generic perceptual naturalness to achieve a massive gain in the specific, task-critical goal of text fidelity (as measured by OCR-A and F1-Score).** The fact that OSEDiff's FID improves while ours slightly degrades, despite our model's vastly superior text reconstruction, is strong evidence that the performance gains are not just from data exposure but from our architecture successfully learning its intended priority.
> > >
> > > We hope this addresses your concerns. We are confident that the corrected pipeline and this clarification will further strengthen the paper, and we thank you again for your diligent review.

---

> > > > ### Comment · Reviewer_AA8X · 2025-08-04
> > > >
> > > > Thank you for the prompt response. My concerns have been fully addressed, and I appreciate the correction of the pipeline in the final version. I will raise my score to **Borderline accept**.

---

> > > > > ### Author Response · Authors · 2025-08-04
> > > > >
> > > > > Thank you very much for your positive feedback and for your engagement throughout this discussion. We are glad that our responses have fully addressed your concerns.
> > > > >
> > > > > We are grateful for your support in raising our score. We confirm that the revised data pipeline and all other clarifications will be carefully integrated into the final version of the paper.
> > > > >
> > > > > Thank you again for your valuable time and insightful review.

---

### Official Review · Reviewer_GxHi · 2025-06-30

**Clarity:** 3
**Significance:** 3
**Originality:** 3
**Rating:** 5
**Confidence:** 4

**Summary:**

This work proposes a novel framework to address the issue that unnatural details of text in image generated by diffusion model, which is practical and  effective.

**Questions:**

Please refer to the weakness.

**Ethical Concerns:**

["NO or VERY MINOR ethics concerns only"]

**Final Justification:**

The proposed text SR method is is practical and  effective. Although there are some limitations in the method design or task formulation, I think authors further improve the quality of the paper and the quality paper still deserves a score of 5.

**Limitations:**

yes

**Paper Formatting Concerns:**

No.

**Quality:**

2

**Strengths And Weaknesses:**

**Strength**:
1. The motivation is strong and novelty. It is an important issue that unnatural details of text in image generated by DM, especially for Chinese character.
2. The design of the devised frame work is simple yet effective. The experimental results demonstrate the effectiveness of the proposed framework, which deliver clear and realistic text in the generated image.

**Weakness**:
1. The confused name of the decoder for segmentation. The joint seg decoder is introduced to generate segmentation map for text in the image. However, it is not used to generate SR image at the same time, therefore, the joint is not proper, which would cause confusion.
2. Unclear explanation in Figure 2. For example, there is no explanation about the orange squares, which could be the LoRA?
3. Confusing descriptions. In line 150-151, why does the $c_{tex}$ need to be located? Isn't it the last one in the $c_y$?
4. Unclear motivation of design for $a_{tex}$. Why are $a_{tex}$ concatenated along the channel dimension and how can it be used to generate the segmentation map solely without $z_H$?
5. It would be better to discuss the parameters, FLOP, and inference time compared to other methods by introducing additional VAE decoder.

---

> ### Author Rebuttal · Authors · 2025-07-31
>
> Thanks for the diligent review and positive feedback on the novelty and design of our work. We appreciate the insightful questions, which help us improve the clarity of our paper. We address each of your points below.
>
> **Q1: The name of the joint seg** **decoder** **is confusing.**
>
> **A1**: Thank you for pointing this out. As mentioned on Line 38 of the main text, "Joint Segmentation Decoders" is an abbreviation for "joint image–text segmentation decoder." To eliminate any potential ambiguity and make this clearer, we will revise the figure to explicitly label the two parallel branches as the "Image Decoder" and the "Text Segmentation Decoder." We apologize for any confusion this may have caused.
>
> **Q2: There is no explanation about the orange squares, which could be the LoRA**?
>
> **A2:** Your assumption is correct; the orange squares indeed represent the LoRA adapters. In Figure 2, we have indicated this with the label "LoRA Adapters" positioned above the U-Net architecture. We adopted this visualization style as it is a common convention, similar to the approach used in prior works such as Image2Image Turbo [1]. We will ensure this label is more prominent in the revised version to avoid any misunderstanding.
>
> **Q3: Why does the** $c_{tex}$ **need to be located?** **Isn't it the last one in the** $c_y$**?**
>
> **A3:** Your understanding is correct for our current implementation. The term "locating" refers to the process of acquiring the specific cross-attention map corresponding to the "text" token. In our current setup with a fixed prompt, this is indeed achieved by simply accessing the token at a known, fixed index.
>
> However, we used the more general term "locating" to maintain broader applicability for future work. This phrasing accommodates potential extensions to dynamic prompts, where the position of the "text" token might vary and would require an explicit search or location step. This choice of terminology was intended to keep the description general rather than tying it to our specific, fixed-prompt implementation.
>
> **Q4: Why are** $a_{tex}$ **concatenated along the channel dimension and how can it be used to generate the** **segmentation** **map solely without** $ z_H$**?**
>
> **A4:** Thank you for this insightful and detailed question. It has two parts, which we will address in order.
>
> 1. **Why Concatenate?** As we mentioned in our preliminary experiments (Line 28), techniques like DAAM have shown that aggregated cross-attention maps hold strong spatial semantic information that can be guided to focus on specific regions (in our case, text). DAAM typically achieves this by upsampling and summing the maps. We reasoned that concatenating these maps along the channel dimension, rather than summing them, would be a more lossless approach. This method preserves the distinct information from each attention layer, creating a rich feature representation that serves as the initial latent input for the text segmentation task.
> 2. **How it works with** $z_H$**:** Crucially, the segmentation map is not generated *solely* from these attention maps. This addresses the second part of your question. Our architecture facilitates interaction with the image latent $z_H$. This is achieved through our proposed Cross-Decoder Interaction Blocks (CDIBs). As detailed in the paper, CDIBs are dual-stream blocks that enable the exchange of features between the image decoding branch (processing $z_H$) and the text segmentation branch (processing $a_{tex}$). This interaction allows the segmentation decoder to be informed by the fine-grained spatial details from $z_H$, leading to a precise and contextually aware final segmentation map.
>
> **Q5: It would be better to discuss the parameters, FLOPs, and inference time compared to other methods by introducing an additional VAE decoder.**
>
> **A5:** Thanks for the suggestion. We provide the table below containing a comparative analysis against standard diffusion model adaptation techniques. All metrics were measured on a single NVIDIA H20 GPU with an input resolution of 512x512 pixels.
>
> | Model Configuration | Extra Trainable Params | FLOPs (GFLOPs) | Inference Time (ms) |
> | ------------------- | ---------------------- | -------------- | ------------------- |
> | SDXL Baseline       | -                      | 2630.49        | 211.19              |
> | SDXL + LoRA         | ~12.61 M               | 2652.47        | 260.85              |
> | SDXL + ControlNet   | ~1263.01 M             | 3006.68        | 260.31              |
> | Ours (TADiSR)       | ~48.29 M               | 4497.96        | 342.32              |
>
> Here is a breakdown of the comparison:
>
> - **Comparison with LoRA:** Our method introduces approximately 35.68M extra trainable parameters beyond a standard LoRA setup, primarily from the new segmentation decoder and the Cross-Decoder Interaction Blocks (CDIBs). While this is a modest increase in parameters, the additional FLOPs and a slight increase in inference time are a direct result of the parallel decoding process. We believe this is a very reasonable trade-off for the substantial improvement in text fidelity, a capability that standard LoRA fine-tuning alone cannot provide.
> - **Comparison with ControlNet:** Our approach is significantly more parameter-efficient. A standard ControlNet adds over a billion parameters by duplicating the U-Net encoder blocks. In contrast, our method achieves fine-grained structural control over text regions with only a fraction of the parameters (~3.8% of a standard ControlNet) and a manageable increase in inference latency. This highlights the efficiency of our joint decoding design for the specific task of text-aware super-resolution.
>
> In summary, while the multi-scale feature interactions within our Joint Segmentation Decoders introduce a certain computational overhead, the resulting increase in latency is acceptable. The reason a large model like ControlNet can be relatively fast is that it operates on a single scale within the latent space. However, this single-scale manipulation is often insufficient for restoring heavily degraded text details, which demands the multi-scale, fine-grained feature processing that our architecture provides.
>
> As this is one of the first works to tackle full-image text super-resolution, our primary focus has been on maximizing model performance and restoration quality. We acknowledge that for practical deployment, further optimization techniques could be applied. We will explore these avenues and discuss them in detail in the next version of our work.
>
> [1] One-Step Image Translation with Text-to-Image Models.

---

> > ### Comment · Reviewer_GxHi · 2025-08-05
> >
> > Thank you to the authors for their response. The authors have addressed my concerns. I will raise my score. But it would be better to clarify the difference of locating and retrieving discussed in Q3.

---

### Official Review · Reviewer_oXY4 · 2025-07-06

**Clarity:** 2
**Significance:** 3
**Originality:** 3
**Rating:** 5
**Confidence:** 5

**Summary:**

The authors propose a Text-Aware Diffusion model for real-world image super-resolution, named TADiSR. Built upon diffusion models, TADiSR integrates Text-Aware Cross-Attention (TACA), a Text Segmentation Decoder (JSD), and Cross-Decoder Interaction Blocks (CDIB) to enhance text extraction and restoration explicitly. The method significantly improves text restoration performance in images.

**Questions:**

See weaknesses.

**Ethical Concerns:**

["NO or VERY MINOR ethics concerns only"]

**Final Justification:**

The proposed text SR method is effective and novel (also supported by other reviewers). Although there are some limitations in the method design or task formulation, I think the paper still deserves a score of 5.

**Limitations:**

The authors have discussed the limitations.

**Quality:**

3

**Strengths And Weaknesses:**

### Strengths

1. TADiSR adopts a dual-branch design to extract text information, thereby enhancing text restoration explicitly. The idea is well-motivated and yields strong performance.
2. The authors also construct a dedicated pipeline for dataset generation, which supports research in this area.
3. Both quantitative and qualitative results demonstrate excellent restoration performance, which is particularly impressive given the tendency of diffusion models to produce hallucinated textures.
4. The ablation study validates the contributions of the JSD and TACA modules, and visual comparisons further support these findings.
5. The paper is well organized. The authors also release code, making the work more solid and reproducible.



### Weaknesses

1. Although the ablation study validates some components, it does not cover the full model. For example, the impact of CDIB is not studied. The study focuses only on network components, while the proposed pipeline is not experimentally evaluated. The authors should validate the effectiveness of the pipeline itself.
2. The pipeline section lacks a flowchart, making it harder to understand. Moreover, pasting text onto other images may not accurately reflect real-world scenarios.
3. In the visual comparisons, the proposed method shows strong performance on Chinese text, but the results of English text are not much different from those of other methods. It is unclear whether this is due to training bias. If other methods like DiffTSR were trained on the same dataset, would the performance gap remain?

---

> ### Author Rebuttal · Authors · 2025-07-31
>
> Thanks for your diligent review and the positive feedback on the innovation and performance of our work. We appreciate the opportunity to address the reviewer's questions and provide further clarification as below.
>
> **Q1: The ablation study should cover the full model. The impact of CDIB should be studied. The authors should validate the effectiveness of the pipeline itself.**
>
> **A1:** Thank you for the suggestion. To maintain the clarity of the main paper, our initial ablation study focused on primary contributions. As requested, we have conducted additional experiments targeting the Cross-Decoder Interaction Block (CDIB) to provide a more comprehensive analysis.
>
> First, we present the quantitative results for these new ablation settings on the FTSR-TE dataset below:
>
> | Settings | PSNR↑ | SSIM↑| LPIPS↓| FID↓  | OCR-A↑ |
> | -------- | --------- | --------- | -------- | ------- | ------- |
> | Exp. 1  | 25.24  | 0.724   | 0.157 | 33.49 | 0.603 |
> | Exp. 2  | 25.15  | 0.722   | 0.159 | 33.81 | 0.595 |
> | Ours    | **25.49** | **0.736** | **0.152** | **32.13** | **0.662** |
>
> Below, we detail the setup and observations for each experiment:
>
> Experiment 1: Disabling Feature Interaction within CDIB
>
> - Setup: In this experiment, we kept the CDIB modules in the architecture but disabled the cross-decoder feature exchange mechanism within them. Each branch (image and segmentation) of the CDIB processed its features independently without receiving features from the other.
> - Observation:  This led to a noticeable degradation in text structure fidelity. While the model could still perform both tasks in parallel, the synergy was lost. As the table shows, all metrics declined compared to our full model. The resulting segmentation masks were less accurate, and text in the super-resolved output appeared blurrier. The OCR-A score dropped significantly, confirming that the interactive feature fusion within CDIB is crucial for the mutual enhancement of the two decoders.
>
> Experiment 2: Removing the CDIB Modules Entirely
>
> - Setup: We removed the CDIB modules from the architecture altogether. The image and segmentation decoders operated as two completely separate, parallel pathways from the latent code, with no points of interaction.
> - Observation: This configuration performed worse than Experiment 1 and approached the performance of the "w/o JSD" variant from our main paper. Without any architectural connection to facilitate information sharing, the model struggled to leverage segmentation priors to guide text reconstruction effectively. This result underscores the necessity of the CDIB module itself as the structural bridge enabling our joint decoding framework to succeed.
>
> **Q2: The pipeline section lacks a flowchart.**
>
> **A2:** Thank you for this valuable suggestion. A flowchart would significantly improve the clarity of the data synthesis pipeline. As it is not feasible to add graphical figures during the rebuttal phase, we have prepared a text-based flowchart below.
>
> ```Plain
> [START]
>    |
>    v
>
>  1. Source Text: CTR Dataset
>     (Text Image Patches with variable quality)
>
>    |
>    v
>
>  2. Super-Resolution via Real-SR Model
>     * Note: To preserve structure, patches are upscaled BEFORE this
>       SR step.
>
>    |
>    v
>
>  3. Quality Control: OCR-based Filtering
>     * Action: Compare text content of original vs. SR patch.
>     * Criteria: Keep only pairs with minimal edit distance to avoid
>       introducing distortions.
>
>    |
>    v
>
>  4. Text Segmentation via SAM-TS Model
>     * Note: This ensures masks are generated for the final, high-
>       quality text, guaranteeing perfect alignment.
>
>    |
>    v
>
>  5. Reliability Control: Further OCR-based filtering to validate
>     the quality and accuracy of the generated segmentation masks.
>
>    |
>    v
>
> The synthesis process combines three distinct data sources:
>
>     (A) High-Quality Text Patches with Aligned Segmentation Masks
>         (The output from Step 5 above)
>
>     (B) Existing Text Segmentation Datasets
>         (To enrich variety)
>
>     (C) Clean, Text-Free Background Images
>         (Derived from the LSDIR Dataset after OCR-based filtering to
>         remove any images containing pre-existing text)
>
>     |
>     | All three sources (A, B, C) are fed into the next step
>     |
>     v
>
>  6. Synthesis Step
>     * Method: Random Pasting or Image Blending
>
>    |
>    v
> [FINAL OUTPUT]
> HR Images with Multiple Text Regions and Paired, High-Fidelity Segmentation Labels.
>
> ```
>
> We will ensure that a clear, professionally drawn flowchart is included in the main body of the paper in the next revision.
>
> **Q3: Pasting text onto other images may not accurately reflect real-world scenarios.**
>
> **A3:** We appreciate the valuable question. The primary motivation for proposing the FTSR dataset was the profound scarcity of high-quality, large-scale datasets with bilingual text scenes suitable for image super-resolution. As far as we know, the available resources are either text recognition datasets with inconsistent image quality (e.g., CTR [1]) or text segmentation datasets with limited scene diversity (e.g., BTS [2]). To facilitate dataset scaling, we adopted this simple yet effective heuristic pasting strategy, which allows us to generate data with greater diversity in scenes and text combinations than is currently available.
>
> It is important to note that, in the short term, the goal of synthetic data is not to perfectly replicate but to approximate real-world data distributions. A model trained on this data can learn the core assumption of our task—the ternary relationship between a low-quality image with text $\textbf{x}_L$, its high-quality counterpart $\textbf{x}_H$, and the corresponding ground-truth segmentation mask $\textbf{s}$, where the mapping from  $\textbf{x}_L$ to $\textbf{x}_H$ must restore sufficient text-structural fidelity to accurately predict $\textbf{s}$.  From this perspective, whether the transition between the pasted text patch and the background is perfectly seamless does not fundamentally alter this core relationship. The primary purpose of this large-scale synthetic data is to reduce the dependency on vast amounts of real-world data and therefore lower data acquisition costs.  However, for practical deployment, a final fine-tuning step on a smaller set of real-world data is always anticipated.
>
> In the long term, the ideal synthetic dataset should indeed simulate the real world as closely as possible to further narrow the gap and reduce the need for real-world fine-tuning. Our current strategy can certainly be improved. We have identified two promising directions for future work:
>
> 1. Image Harmonization: Employing techniques like Poisson blending to create a smoother, more natural transition between the text patches and the background images.
> 2. Text Image Rendering based on Generative Models: Using text-to-image models with strong text rendering capabilities (e.g., JoyType [3], AnyText [4], Seedream [5]) to generate entire scenes.
>
> We have conducted preliminary experiments based on both of these approaches:
>
> By applying Poisson blending to a subset of our training data, we observed a marginal improvement in perceptual quality, although the impact on quantitative metrics like PSNR and OCR-A was minimal. This suggests that while harmonization can enhance visual appeal, our core pipeline already provides the essential structural information needed for effective training.
>
> Regarding generative text image rendering methods, our specific experimental findings are as follows:
>
> - JoyType relies on manually rendered text outlines, which makes it challenging to scale for large-scale dataset generation.
> - AnyText requires manual specification of text placement and frequently produces instances with missing or distorted strokes, or text that does not strictly adhere to the prompt. This limits its practical applicability for high-quality ground-truth generation.
> - Seedream 3.0 is more promising, as it can generate high-resolution, bilingual images that largely conform to text prompts without requiring manual text placement. However, in our experiments, we still encountered issues such as text positioning inaccuracies relative to the prompt, structural distortions in small characters, and sharpening artifacts at text edges. These issues could potentially compromise the naturalness and fidelity of the text regions in full-image super-resolution results.
>
> In summary, current generative text image rendering models are still some distance from consistently producing high-quality, full-image text ground-truths suitable for large-scale SR training. We will add the discussion in the next version of our paper to thoroughly evaluate the applicability of different data synthesis pipelines for full-image text super-resolution data generation.
>
> Thank you again for your constructive feedback. We hope our responses have addressed your concerns.
>
> **References**:
>
> [1] Benchmarking Chinese Text Recognition: Datasets, Baselines, and an Empirical Study.
>
> [2] A Bi-Lingual Benchmark for Text Segmentation in the Wild.
>
> [3] JoyType: A Robust Design for Multilingual Visual Text Creation.
>
> [4] AnyText: Multilingual Visual Text Generation And Editing.
>
> [5] Seedream 3.0 Technical Report.

---

> > ### Comment · Reviewer_oXY4 · 2025-08-04
> >
> > Thank you to the authors for their response. The authors have addressed my concerns. I will raise my score.

---

### Comment · Area_Chair_Zpqg · 2025-08-03
**Discussions for Paper #2436**

Dear Reviewers,

Thanks for serving as reviewers for NeurIPS 2025! The author-reviewer discussion phase is open until August 6, 11:59 PM AoE.

This paper receives 2xBorderlineReject and 2xBorderlineAccept. The authors have provided a rebuttal.

Please kindly read the rebuttal and the other reviews at your earliest convenience. Please check whether your main concerns have been addressed.

Thanks,

Your AC

---

### Note · Authors · 2025-08-15

Dear Area Chair and Reviewers,

We are deeply grateful for the thorough and constructive feedback provided by all reviewers. The interactive discussion period has been invaluable, allowing us to clarify key aspects of our work and significantly strengthen our paper.

We are pleased that our detailed responses, including additional ablation studies (for CDIB), a text-based flowchart of our data pipeline, and new experiments on higher-scale (×8) degradation, have successfully addressed all initial concerns. The reviewers' feedback has led to a more robust validation of our method's novelty, performance, and the general utility of our FTSR dataset.

Specifically, we have demonstrated that:
1.  Our architectural components, including the Cross-Decoder Interaction Blocks (CDIB), are crucial for performance.
2.  Our data synthesis pipeline is sound, and we have clarified the quality control steps that ensure data fidelity.
3.  The performance gains of TADiSR stem from our novel text-aware architecture, not just data exposure, as evidenced by comparisons where baselines were also trained on our FTSR dataset.
4.  Our method maintains a significant advantage even under more severe (×8) degradation against strong baselines like HAT.

All reviewers (oXY4, GxHi, AA8X, J5Zt) have acknowledged that their concerns were fully addressed and have consequently raised their scores. We are confident that the final version of our paper, incorporating all the clarifications and new results prompted by this discussion, will offer unique insights into text-aware image restoration. We hope our work will inspire further attention in this important area and contribute to the continued development of diffusion models.

Thank you once again for your time and diligent consideration.

---

### Decision · Program_Chairs · 2025-09-17

**Decision:**

Accept (poster)

**Comment:**

This paper proposes a text-aware diffusion model and the FTSR dataset for real-world image super-resolution (SR), effectively preserving the structural fidelity of text regions. The submission received 4 positive ratings (2 Accept and 2 Borderline Accept), and all reviewers agreed that the work is both effective and novel.

The AC finds that the paper offers helpful insights into text-aware image restoration, particularly in preserving text structure fidelity, which has been a long-standing challenge for diffusion-based SR methods. The authors are encouraged to incorporate the reviewers’ constructive feedback in the final version, such as clarifying the distinction between “locating” and “retrieving” raised in [Q3] by Reviewer GxHi, and correcting the data pipeline issue highlighted by Reviewer AA8X. The release of the FTSR dataset is particularly encouraged, as it is expected to facilitate further progress in this field. The AC recommends acceptance of this paper.